# Neuropilin 1 and its inhibitory ligand mini-tryptophanyl-tRNA synthetase inversely regulate VE-cadherin turnover and vascular permeability

Noemi Gioelli [1,2], Lisa J. Neilson[3], Na Wei[4], Giulia Villari [1,2], Wenqian Chen[4], Bernhard Kuhle [4], Manuel Ehling[5,6], Federica Maione[1,2], Sander Willox [5,6], Serena Brundu [2,7], Daniele Avanzato[1,2], Grigorios Koulouras [3], Massimiliano Mazzone [5,6,7,8,9], Enrico Giraudo [2,7], Xiang-Lei Yang[4], Donatella Valdembri [1,2], Sara Zanivan [3,10 ✉] & Guido Serini [1,2 ✉]

The formation of a functional blood vessel network relies on the ability of endothelial cells (ECs) to dynamically rearrange their adhesive contacts in response to blood flow and guidance cues, such as vascular endothelial growth factor-A (VEGF-A) and class 3 semaphorins (SEMA3s). Neuropilin 1 (NRP1) is essential for blood vessel development, independently of its ligands VEGF-A and SEMA3, through poorly understood mechanisms. Grounding on unbiased proteomic analysis, we report here that NRP1 acts as an endocytic chaperone primarily for adhesion receptors on the surface of unstimulated ECs. NRP1 localizes at adherens junctions (AJs) where, interacting with VE-cadherin, promotes its basal internalization-dependent turnover and favors vascular permeability initiated by histamine in both cultured ECs and mice. We identify a splice variant of tryptophanyl-tRNA synthetase (mini-WARS) as an unconventionally secreted extracellular inhibitory ligand of NRP1 that, by stabilizing it at the AJs, slows down both VE-cadherin turnover and histamine-elicited endothelial leakage. Thus, our work shows a role for NRP1 as a major regulator of AJs plasticity and reveals how mini-WARS acts as a physiological NRP1 inhibitory ligand in the control of VE-cadherin endocytic turnover and vascular permeability.

[1] Department of Oncology, University of Torino School of Medicine, Candiolo (TO), Italy. [2] Candiolo Cancer Institute - Fondazione del Piemonte per l'Oncologia (FPO) Istituto di Ricovero e Cura a Carattere Scientifico (IRCCS), Candiolo (TO), Italy. [3] Cancer Research UK Beatson Institute, Glasgow, UK. [4] Department of Molecular Medicine, The Scripps Research Institute, La Jolla, CA 92037, USA. [5] Center for Cancer Biology, Department of Oncology, University of Leuven, Leuven 3000, Belgium. [6] Center for Cancer Biology, VIB, Leuven 3000, Belgium. [7] Department of Science and Drug Technology, University of Torino, Torino, Italy. [8] Molecular Biotechnology Center (MBC), University of Torino, Torino, Italy. [9] Department of Molecular Biotechnology and Health Sciences, University of Torino, Torino, Italy. [10] Institute of Cancer Sciences, University of Glasgow, Glasgow, UK. ✉email: S.Zanivan@beatson.gla.ac.uk; guido.serini@ircc.it

During evolution, the *Neuropilin-1* (*NRP1*) gene first emerged in vertebrates concomitantly with the appearance of the cardiovascular apparatus and a more elaborated nervous system[1]. Consistently, during embryonic development, the expression of NRP1 transmembrane protein is enriched in neurons and vascular endothelial cells (ECs)[2]. Furthermore, improper cardiac outflow tract (OFT) septation[3], along with severe abnormalities of blood vessels[3] and peripheral nerves[4], are observed in *Nrp1* knockout mouse embryos, revealing key morphogenetic functions of NRP1 receptor during patterning of vertebrate body plan. Yet, differently from neurons[5], the molecular mechanisms by which NRP1 signals in ECs to control vascular morphogenesis are still poorly understood[6].

The extracellular coagulation factor V/VIII homology domain b1 of NRP1 binds with high affinity to the C-terminal arginine of chemorepulsive secreted class 3 semaphorin proteins (SEMA3s), such as SEMA3A or SEMA3C, and the angiogenic vascular endothelial growth factor-A (VEGF-A)[7]. For this reason, NRP1 is mainly postulated to function as a co-receptor of SEMA3s and VEGF-A ligands, that signal via type A/D plexin (PLXNA/D) receptors[6,8,9] and VEGF receptor-2 (VEGFR-2)[10,11]. In fact, neuronal guidance[4] and OFT septation[12] defects caused by *Nrp1* gene deletion are largely phenocopied in mice lacking *Sema3a*[13] and *Sema3c*[14], respectively. However, different vascular abnormalities are phenotypically observed upon murine *Nrp1*[3,15] or *Sema3a*[16,17] or *Sema3c*[14] knock-out. Moreover, mice harboring a point mutation disrupting the VEGF-A binding to NRP1 survive to adulthood with normal vasculature[18]. Altogether, these data hint that in ECs NRP1 may signal and play a functional role(s) without requiring the binding with extracellular ligands. In this respect, even if, as conceivable, VEGF-A and SEMA3A promote NRP1 internalization[19], we have previously shown that, in ECs, NRP1 displays the ligand-independent ability to associate with and promoting the endocytosis of active α5β1 integrin[20], the main fibronectin (FN) receptor driving embryonic vascular development and cancer angiogenesis[21], thanks to the interaction of its short cytosolic tail with trafficking adaptors, such as GIPC1. In addition, the b1 domain-binding carboxy (C)-terminal basic sequence (C-end Rule or CendR) motif-containing peptides are taken up by NRP1-dependent internalization[22,23] and cause NRP1-driven blood vessel leakage, similarly to VEGF-A[22,24]. Our recent finding that high-affinity binding to NRP1 is required for SEMA3A to increase integrin internalization and to elicit vascular permeability[25] further suggests that NRP1 displays autonomous function(s), such as the constitutive endocytosis of interacting membrane proteins[20], and that may be modulated by b1 domain-interacting ligands. Interestingly, the secreted glycyl tRNA synthetase (GARS) mutants implicated in type 2D Charcot-Marie-Tooth (CMT2D) neuropathy compete with VEGF-A for the binding to the extracellular b1 domain of NRP1, which is a CMT2D modifier gene[26,27]. Indeed, during evolution aminoacyl-tRNA synthetase (aaRS) proteins have acquired additional domains and motifs that endowed them with nontranslational properties, such as those of being secreted and acting as extracellular proteins that signal through mechanisms that are still incompletely characterized[27–29]. For example, vertebrate tryptophanyl-tRNA synthetase (WARS) has acquired, passing from fish to humans, a N-terminal WHEP domain to negatively regulate its anti-angiogenic activity[30].

To draw a detailed molecular portrait of the protein interaction network potentially responsible for NRP1 intrinsic biological activities, we have analyzed by quantitative mass spectrometry (MS) its interacting partners both on the surface and in the endosomal compartment of ECs in the absence of exogenously added ligands. Grounding on our unbiased comprehensive analysis, we revealed that, in ECs, NRP1 interacts with and promotes the endocytic turnover of VE-cadherin at cell-to-cell contacts,

thus allowing histamine to elicit vascular permeability. Furthermore, we identified NRP1 as a potential receptor for several aaRS family members. Among those, an unconventionally secreted splice variant of WARS without the WHEP domain (mini-WARS), upon binding the extracellular region of NRP1, decreases both NRP1 and VE-cadherin endocytosis at adherens junctions (AJs) and impairs EC permeability in response to histamine in a NRP1-dependent manner.

## Results

**A proteome-scale map of the NRP1 interactome in unstimulated ECs.** To get unbiased insights on the molecular networks through which NRP1 controls EC and blood vessel functions independently from its ligands, we implemented a modified haloalkane dehalogenase HaloTag (HT)-based approach, which enables effective surface labeling and pull-down of proteins[31], and combined it with quantitative mass spectrometry (MS) analysis. To this aim, we generated a NRP1 construct containing a N-terminal modified HT that covalently binds membrane-impermeable chloroalkane containing ligands, such as biotin, thus allowing an effective antibody (Ab)-independent isolation of HT-tagged proteins on streptavidin resin[31]. We subcloned in a lentiviral vector a construct in which the HT encoding sequence was fused immediately 3' to that of the signal peptide of silencing-resistant murine Nrp1 (HT-NRP1) (Fig. 1a). We transduced human ECs with HT-NRP1 (HT-NRP1-ECs) and verified that the corresponding protein was expressed at constant levels for 10 days by western blot analysis (Fig. 1b). Confocal fluorescent microscopy showed that, upon 15 min incubation at 4 °C (a temperature known to inhibit endocytosis), a membrane-impermeable Alexa 660-labeled chloroalkane ligand evenly distributed on the surface of unstimulated HT-NRP1-ECs (Fig. 1c). Moreover, the ligand localized in early endosome antigen 1 (EEA1) positive early endosomes when HT-NRP1-ECs were transferred at 37 °C for 3 min to allow internalization (Fig. 1d). Next, HT-NRP1-ECs were incubated with a membrane-impermeable biotin-labeled chloroalkane (HT-PEG-Biotin) ligand and streptavidin beads used to pull-down biotinylated proteins from their total lysate. VEGF receptor-2 (VEGFR-2) was co-precipitated with HT-NRP1 and stimulation with VEGF-A 165 increased their association, as expected for its endogenous counterpart[16,32] (Fig. 1e). We have previously demonstrated that NRP1 associates with and promotes the internalization of active α5β1 integrin when ECs spread and adhere on FN[20]. We found that, similarly to wild type NRP1[20], lentivirus-delivered HT-NRP1 successfully rescues the defective adhesion to FN of NRP1 silenced (siNRP1) ECs (Fig. 1f). Thus, the N-terminal tagging with HT does not impair NRP1 ability of reaching EC surface, being trafficked through endosomal compartments, and playing physiological roles, such as increasing its association with VEGFR-2 upon VEGF-A stimulation or promoting α5β1 integrin-mediated adhesion on FN (Fig. 2a).

Then, since we and others reported that NRP1 acts as an endocytic receptor[20,22,23,33] (e.g., for integrins, Fig. 2a), we sought to identify proteins that in basal conditions, directly or indirectly, associate with NRP1 either at the EC plasma membrane or when internalized in the endosomes, using label-free quantitative MS proteomics (Fig. 2a). Non-stimulated HT-NRP1 ECs were kept in basal conditions in the absence of exogenous ligands and surface labeled or not (as control, Ctrl) with membrane-impermeable HT-PEG-Biotin ligand at 4 °C. Furthermore, cells were incubated at 37 °C for 3 min to allow endocytosis and catch its interacting partners also in endosomal compartments. We isolated the NRP1 surface interactome from ECs that were kept at 4 °C, and that from cells incubated at 37 °C for 3 min (referred to as

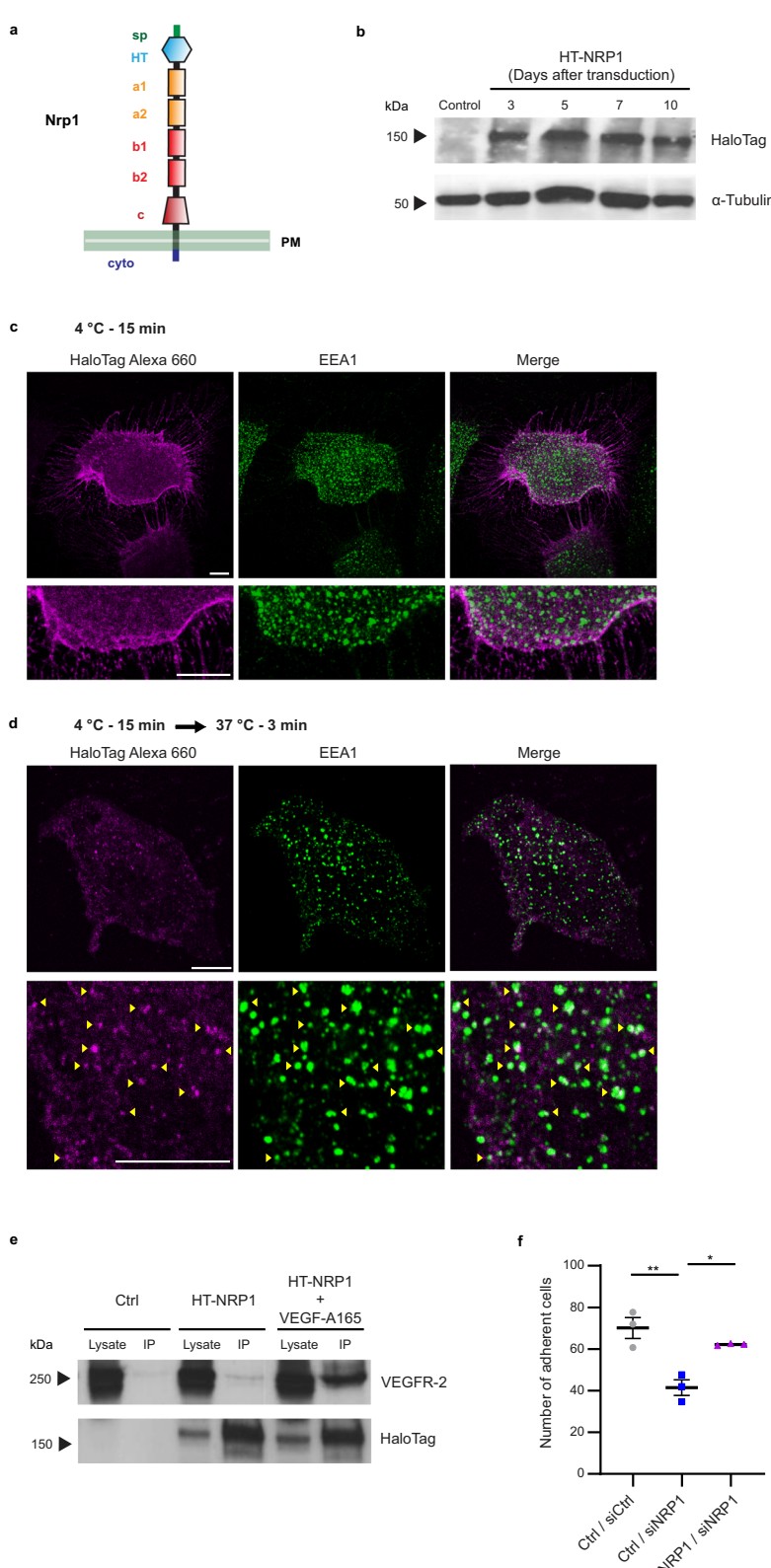

endosomal interactome), although the latter contained both endosomal and surface interactors, as only a proportion of NRP1 was endocytosed. We performed MS proteomic analysis of four independent experiments and used the MaxQuant label-free quantification (LFQ) algorithm for protein quantification[34]. Principal component analysis (PCA) of the 966 quantified proteins identified components that segregated the three

experimental conditions (Supplementary Fig. 1a), as indication that our strategy may have successfully identified NRP1 interactors. Next, we performed a t-test analysis, which identified 114 distinct putative direct or indirect interaction partners of NRP1 (see "Methods"). Of those potential NRP1 interactors, 61 proteins represented the NRP1 surface interactome and 86 the NRP1 endosomal interactome (Supplementary Data 1). Of note,

**Fig. 1 Generation and expression of a functional HaloTag-NRP1 construct in ECs. a** HaloTag protein tag fused at the N-terminus of mouse NRP1 (HT-NRP1); sp, signal peptide; HT, HaloTag; a1/a2, complement C1r/C1s, Uegf, Bmp1 (CUB) domain 1/2; b1/b2, coagulation factor VIII/V (FVIII/FV) domain 1/2; c, meprin/A5/µ-phosphatase (MAM) domain. **b** Western blot analysis of exogenous HT-NRP1 construct and endogenous α-Tubulin in ECs transduced with empty pCCL lentivirus (Ctrl) or carrying HT-NRP1 construct. A representative experiment out of three is shown. **c, d** HaloTag NRP1 localizes at the cell membrane when endocytosis is inhibited, whereas it co-localizes with the EEA1 endocytic marker during internalization. Fluorescent confocal microscopy on ECs transduced with pCCL HT-NRP1, incubated for 15 min at 4 °C (to inhibit endocytosis) with the cell impermeable HaloTag-Alexa 660-labeled chloroalkane (magenta) fluorescent ligand (**c**), and then allowed to internalize membrane proteins by a shift to 37 °C (**d**). ECs were simultaneously stained for the endogenous early endosomal marker EEA1 (green). HaloTag-Alexa 660 bound HT-NRP1 localized on the surface of ECs kept at 4 °C (**c**) and is efficiently endocytosed only upon incubation at 37 °C for 3 min, where HT-NRP1 co-localized in EEA1-positive early endosomes (**d** yellow arrows). Bottom panels are magnifications of top panels. Scale bar is 10 µm. A representative experiment out of three is shown. **e** VEGF-A165 promotes the association of HT-NRP1 to VEGFR-2. Empty pCCL (Ctrl) or pCCL-HaloTag-NRP1 transduced ECs (HT-NRP1) were labeled with 3 µM HaloTag cell impermeable Biotin ligand for 15 min at 4 °C (to inhibit endocytosis) and stimulated or not with VEGF-A165 (30 ng/ml, 10 min at 37 °C), then immunoprecipitated on streptavidin-beads. A representative experiment out of two is shown. **f** HT-NRP1 rescues the defective adhesion of siNRP1 ECs to fibronectin. Results are the mean of three independent experiments (each in technical triplicate) ±SEM. Statistical analysis: parametric one-way analysis of variance (ANOVA) with Bonferroni correction; *P value = 0.0199, **P value = 0.0041. Source data are provided as Source Data file.

the surface and the endosomal interactomes shared more than 50% hits (33 proteins). Intriguingly, we could not find any proteins which was enriched at a similar extent as NRP1 (Supplementary Fig. 1b), suggesting that NRP1 interactome relies on transient rather than constitutive interactions, in agreement with the fact that NRP1 has been found to participate and modulate several different vascular signaling pathways[11].

To understand the roles that NRP1 may play in regulating receptor functions through endosomal trafficking, we first extracted from our list of 114 putative NRP1 interactors all the transmembrane proteins (based on their annotated topology in UniProt[35], Fig. 2a, b) and those associated to the endosomal compartment (based on the subcellular location defined in UniProt, Fig. 2a and Supplementary Fig. 2a). We found 16 transmembrane proteins, which encompassed "membrane", "cell membrane", "focal adhesion" and "cell junction" proteins (Fig. 2b), and 23 endosomal proteins that localize at the "lysosome", "early endosome" "late endosome" and "cytoplasmic vesicle" compartments (Supplementary Fig. 2a). In agreement with previous findings[20,36], the α5, αv, and α2 integrin subunits were enriched on both EC surface and endosome interactome of NRP1 (Fig. 2a, b and Supplementary Data 1). NRP1 also associated with tetraspanin CD63, which interacts with integrins and promotes adhesion to the ECM[37]. Of note, we identified several intercellular adhesion receptors, namely VE-cadherin (CDH5), PECAM1 (CD31), and MCAM (CD146), as potential NRP1 transmembrane partners. In sum, one-third of the transmembrane proteins that we found to associate with NRP1 mediate or regulate EC adhesion (Fig. 2a, b and Supplementary Data 1).

Next, we used STRING to analyze the remaining candidate NRP1 interactors. Category enrichment analysis based on KEGG annotations found that the significantly enriched categories were mostly related to cell adhesion, infection, and cell metabolism (Supplementary Fig. 2b–d and Supplementary Data 2). Intriguingly, our analysis also highlighted a substantial enrichment of proteins annotated to "Aminoacyl-tRNA biosynthesis", in particular WARS1, GARS1 and HARS1 over the twenty members of the cytoplasmic aaRS family (Fig. 2c).

In sum, our proteomic analysis has identified established interactors of NRP1, but also a plethora of previously unknown putative interactors. These encompass proteins involved in mechanisms known to be regulated by NRP1, such as adhesion receptors, aaRSs, immune and metabolic regulators, but also proteins involved in processes in which NRP1 role has not been yet described, notably glycolysis. Hence, we propose our NRP1 interactome as a comprehensive map of known and unknown NRP1 interactors in ECs.

**NRP1 localizes at intercellular contacts and interacts with cell-to-cell adhesion receptors.** We[20] and others[36,38–40] have previously shown that NRP1 localizes at ECM adhesions where it interacts with integrins and regulates their function. Our MS analysis unveiled that, in addition to integrins and their cytosolic interactors, NRP1 may also associate with cell-to-cell adhesion receptors, such as VE-cadherin (CDH5) and platelet EC adhesion molecule 1 (PECAM1/CD31). Indeed, cell-to-cell and cell-to-ECM contacts share different adaptor and signaling proteins, and their mechanical crosstalk is critical for FN fibrillogenesis and directional collective cell migration[41]. Thus, NRP1 may represent a regulatory protein that functions as an endocytic receptor both at ECM and intercellular adhesion sites. To verify this hypothesis, we determined NRP1 subcellular positioning relatively to intercellular adhesion molecules identified in our MS interactome. Confocal microscopy analysis showed that in unstimulated ECs in culture, NRP1 clearly co-localizes at cell-to-cell contacts with VE-cadherin (Fig. 3a) and PECAM1 (Fig. 3b), similarly to what recently reported in vivo in quiescent adult mouse dermis blood vessels[42]. Next, to further confirm our MS analyses, we assessed whether NRP1 may physically associate with such cell-to-cell adhesion receptors. Unstimulated ECs were live incubated with a mouse mAb recognizing the extracellular portion of NRP1 or with non-immune IgGs (as control), lysed, immunoprecipitated, and analyzed by western blotting with mAbs recognizing either VE-cadherin or PECAM1. NRP1 co-immunoprecipitated with VE-cadherin (Fig. 3c) and PECAM1 (Fig. 3d). Furthermore, in vitro pull-down assay employing an agarose-conjugated anti-NRP1 antibody confirmed that the purified Fc-tagged recombinant whole extracellular portions of VE-cadherin and NRP1 directly interact (Fig. 3e). Hence, in ECs NRP1 localizes at cell-to-cell contacts and it interacts with different intercellular adhesive receptors, such as VE-cadherin, independently from exogenous VEGF-A stimulation.

**NRP1 promotes VE-cadherin turnover and histamine-elicited endothelial permeability.** The contact of NRP1 with active integrins fosters their internalization and turnover at ECM adhesions. Therefore, we investigated whether NRP1, in addition to co-localize and directly interact with VE-cadherin, also controls its turnover at AJs. ECs were transfected with either a pool of three different small interfering RNAs (siRNAs) targeting human NRP1 (siNRP1), which we have previously validated, or control non-targeting siRNA (siCtrl)[20,43]. Fluorescent confocal microscopy revealed that, compared to siCtrl ECs, siNRP1 ECs display larger VE-cadherin-containing AJs whose mean fluorescence intensity (MFI) was significantly higher (1.8-fold ± 0.1%, P < 0.001) (Fig. 4a). Furthermore, the transduction of a silencing-resistant mouse Nrp1

**a**

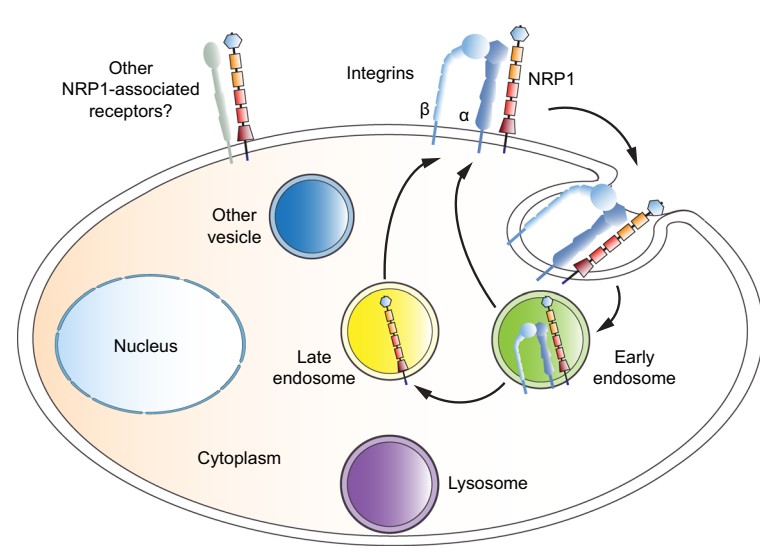

**b**    Transmembrane NRP1 interactors

**c**    Aminoacyl-tRNA biosynthesis

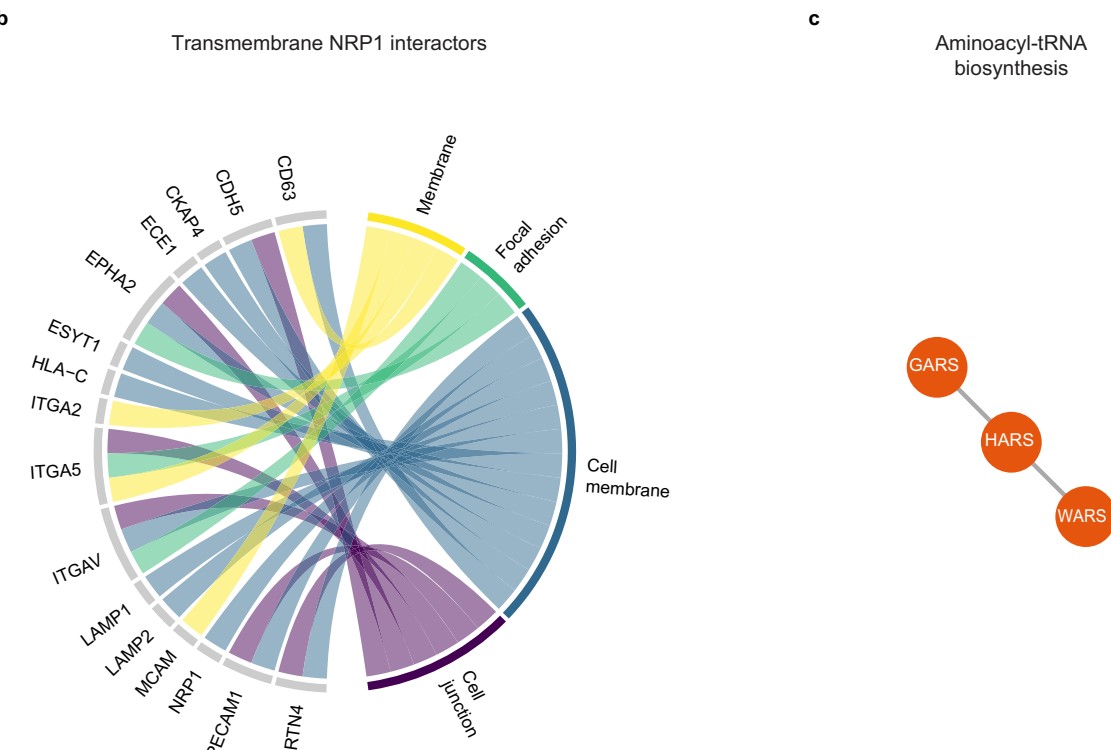

**Fig. 2 NRP1 interactome. a** Schematic drawing summarizing the role of NRP1 as a pro-endocytic receptor[20,22,23,33] that promotes the internalization and recycling of associated plasma membrane receptors, such as integrins[20,33] and other still to be identified receptors (in gray). **b** Cell adhesion receptors as preferential potential transmembrane NRP1 interactors. Chord diagram of transmembrane proteins found to interact with NRP1. Each protein was assigned to one or more subcellular locations based on UniProt annotation. In addition, to associate with transmembrane proteins mediating or regulating EC adhesion, NRP1 also associated with the endoplasmic reticulum reticulon 4 (RTN4) transmembrane protein that promotes cell adhesion by regulating integrin traffic[101]. Moreover, our identification of the transmembrane metallopeptidase endothelin converting enzyme-1 (ECE1) as a NRP1 partner on the EC surface may suggest that the TBX1-independent control of cardiac OFT morphogenesis by endothelial NRP1[102] may involve ECE1[103,104], which functions by degrading receptor-bound extracellular ligands upon endocytosis (Supplementary Data 1). **c** Aminoacyl-tRNA proteins were found to be significantly enriched in the NRP1 interactome, after excluding transmembrane (**b**) and endosomal (Supplementary Fig. 2a) NRP1 interactors. STRING was used to perform enrichment analysis of KEGG pathways and to determine the physical and functional protein-protein interactions that define the edges between nodes. The plots was generated with R. Source data are provided as a Supplementary Data file.

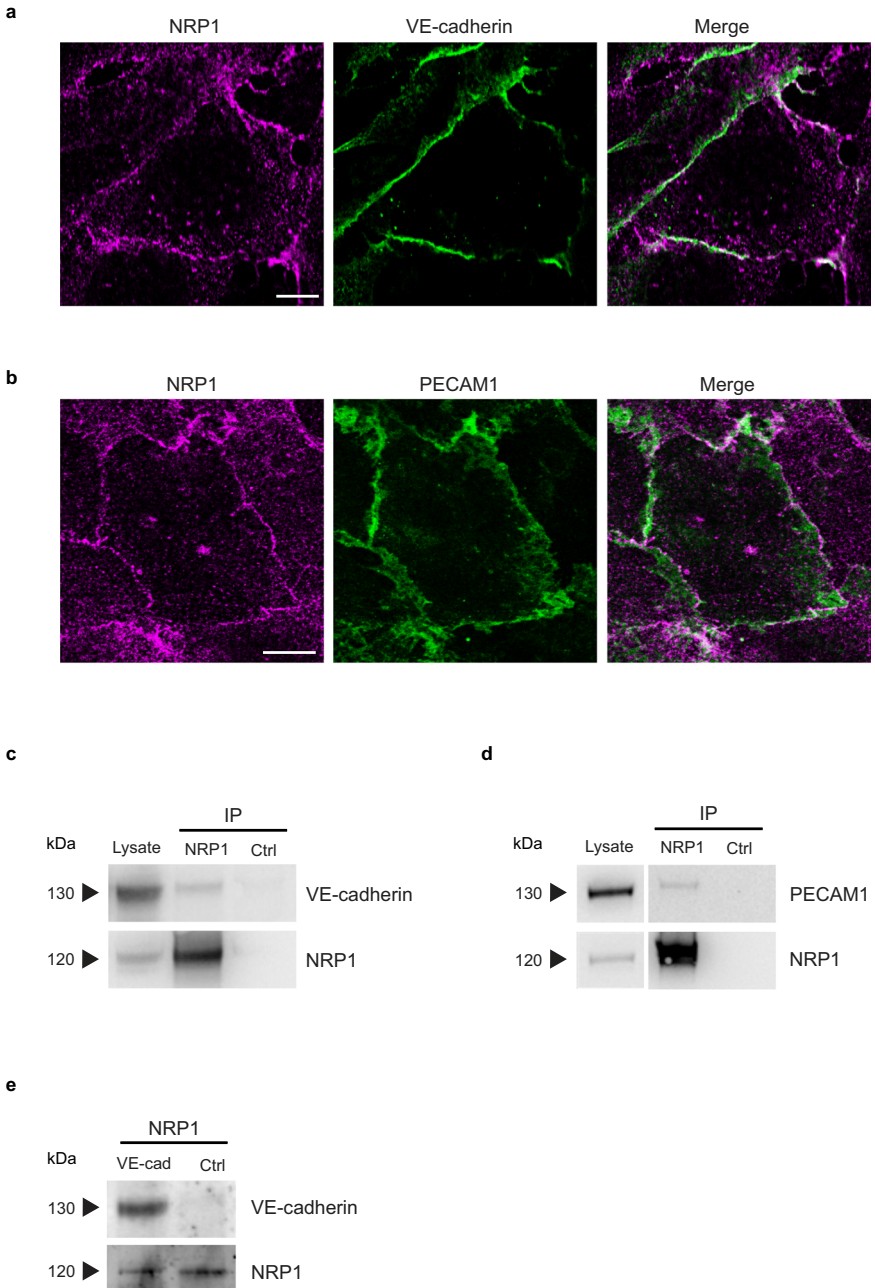

**Fig. 3 Endogenous NRP1 localizes at intercellular contacts and interacts with cell adhesion receptors in ECs. a, b** NRP1 co-localizes with VE-cadherin and PECAM1 at endothelial intercellular contacts. Confocal microscopy analysis of endogenous NRP1 (magenta) and VE-cadherin (**a**) and PECAM1 (**b**) (green) in cultured ECs. Scale bar is 10 μm. **c, d** NRP1 co-immunoprecipitates with VE-cadherin and PECAM1 intercellular adhesion receptors. Co-immunoprecipitation of endogenous NRP1 and VE-cadherin (**c**) and PECAM1 (**d**) from lysates of cultured ECs that were previously incubated live with the mouse mAb (R&D Systems—MAB3870) recognizing the extracellular domain of human NRP1 or control mouse IgGs (Ctrl). A representative experiment out of five (**c**) or four (**d**) is shown. **e** Pull-down experiments further confirm VE-cadherin as a NRP1 binding partner. Representative western blot analysis (out of three biological replicates) of in vitro pull-down assay in which NRP1-Fc-tagged bound to an agarose-conjugated anti-NRP1 antibody was incubated with recombinant VE-cadherin-Fc-tagged or Fc alone. Source data are provided as Source Data file.

construct, which we previously characterized[20] (Fig. 4b), effectively rescued the abnormal accumulation of VE-cadherin at siNRP1 EC AJs (Fig. 4a). Notably, NRP1 silencing did not alter total VE-cadherin protein amounts, as evaluated by western blot (Fig. 4b). Next, we transduced and quantified by fluorescence recovery after photobleaching (FRAP) the turnover of VE-cadherin-mCherry[44] at AJs in siNRP1 ECs compared to siCtrl ECs. FRAP analysis showed that VE-cadherin-mCherry mobile fraction was reduced by $37 \pm 0.4\%$ upon NRP1 knockdown in ECs (Fig. 4c). Altogether

these data indicate that in ECs NRP1 physically interacts with VE-cadherin, negatively regulating its accumulation and promoting its turnover at AJs.

Similar to integrin-containing ECM adhesions[33,43,45–48], the turnover of AJs largely impinges on cycles of cadherin endocytosis and recycling back to the plasma membrane[49,50]. Hence, we sought to biochemically gauge the effect of NRP1 silencing on VE-cadherin internalization in ECs by implementing a well-established method employed to quantitatively monitor integrin endocytosis[20,51–53].

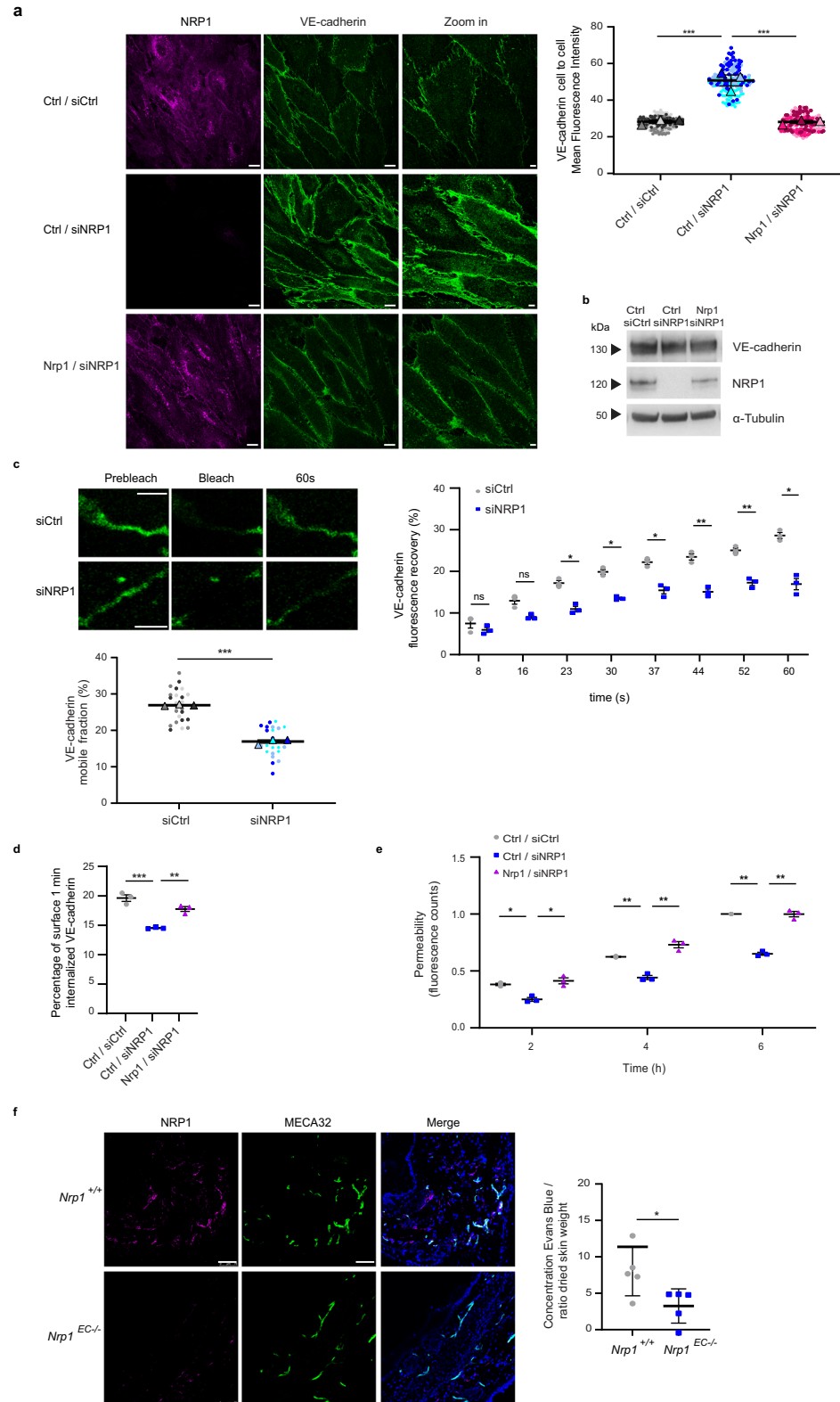

Confluent ECs were surface labeled with cleavable biotin at 4 °C and then incubated at 37 °C for 1 min to allow internalization of biotinylated plasma membrane proteins. Next, the biotin remaining on cell-surface proteins was cleaved and VE-cadherin endocytosis quantified by capture ELISA on cell lysates, employing an anti-VE-cadherin Ab as a catcher. Knockdown of NRP1 diminished by 26 ± 0.004% the quantity of VE-cadherin internalized by ECs and

this endocytic defect was rescued by the transduction of silencing-resistant mouse Nrp1 (Fig. 4d). Conversely, the lack of NRP1 did not affect the endocytosis of transferrin receptor 1 (Supplementary Fig. 3a), a model cargo for receptor trafficking studies[54] that does not interact with NRP1 (Fig. 2b and Source Data 1).

VE-cadherin endocytosis is one of the key processes through which VEGF-A and vascular permeability-inducing

**Fig. 4 NRP1 promotes VE-cadherin turnover and histamine-elicited endothelial permeability. a** NRP1 impairs VE-cadherin localization at endothelial cell-to-cell contacts. Confocal microscopy on control (siCtrl) or NRP1 (siNRP1) silenced ECs rescued with empty vector (Ctrl/siCtrl and Ctrl/siNRP1) or with mouse Nrp1 (Nrp1/siNRP1) and stained for NRP1 (magenta) and VE-cadherin (green) and quantification of VE-cadherin fluorescence intensity in AJs. Scale bar 10 µm. Mean ± SEM of three independent experiments (depicted in different colors). One-way ANOVA with Bonferroni correction; ***P value = 0.0004.
**b** NRP1 silencing does not affect VE-cadherin protein levels. A representative experiment of three is shown. **c** FRAP analysis reveals NRP1 promotes VE-cadherin turnover at AJs in ECs transduced with VE-cadherin-mCherry and silenced or not for NRP1. Above, VE-cadherin fluorescence recovery = normalized signal in each cell before and after photobleaching to 100% and 0%. Three independent experiments (8 cells each) mean ± SEM is shown. Below, mobile fraction = fluorescence recovered post-bleach —after photobleaching/pre-bleach ROI fluorescence—after photobleaching. Mean of three independent experiments ± SEM is shown. Scale bar 5 µm. Two-way ANOVA with Bonferroni correction; ns = P value ≥ 0.05, *P value ≤ 0.05, **P value ≤ 0.01, ***P value = 0.000028. **d** NRP1 promotes VE-cadherin internalization. VE-cadherin endocytosis in Ctrl/siCtrl, Ctrl/siNRP1 or Nrp1/siNRP1 ECs measured by capture ELISA. Mean ± SEM of three independent experiments is shown. One-way ANOVA with Bonferroni correction; **P value = 0.0037, ***P value = 0.0003. **e** NRP1 sustains histamine-elicited endothelial permeability. Quantification of Dextran through Ctrl/siCtrl, Ctrl/siNRP1 or Nrp1/siNRP1 EC monolayer after histamine. Mean ± SEM of three independent experiments is shown. Two-way ANOVA with Bonferroni correction; *P value ≤ 0.05, **P value ≤ 0.01. **f** Vascular permeability reduction in Nrp1[EC−/−] mice. Left, confocal microscopy on Nrp1[+/+] or Nrp1[EC−/−] mice ear sections stained for NRP1 (magenta), MECA32 (green), and DAPI (blue). Scale bar 75 µm. Right, histamine permeability in Nrp1[+/+] or Nrp1[EC−/−] mice (Evans Blue/skin weight). Data are mean ± SD (n = 5 per group). Two-tailed heteroscedastic Student's t test; *P value = 0.0321. Source data are provided as Source Data file.

small molecules, such as histamine and bradykinin, trigger AJ dissolution, endothelial barrier opening, and permeability[50,55,56]. It has been reported that NRP1 enables its ligand VEGF-A to stimulate vascular permeability[24,42]. Yet, as shown above, in ECs NRP1 is autonomously able to promote VE-cadherin turnover and internalization in the absence of exogenously added VEGF-A (Fig. 4a–d). To assess the effect of NRP1 knockdown in VEGF-A-independent vascular permeability, we used histamine, which signals via G-protein coupled histamine receptor H1 in ECs[57]. In vitro time course transendothelial flux tracer analyses[58] demonstrated that histamine-elicited vascular permeability was clearly diminished in siNRP1 ECs compared with siCtrl ECs and this permeability defect was rescued by the transduction of silencing-resistant murine Nrp1 (Fig. 4e). Moreover, also in vivo EC-specific knock-out of Nrp1 in mice resulted in a significant reduction of histamine-induced vascular permeability (Fig. 4f), without affecting blood vessel density (Supplementary Fig. 3b). Hence, NRP1 localizes at EC AJs where it interacts and promotes the endocytic turnover of VE-cadherin, thus favoring histamine-stimulated vascular permeability both in vitro and in vivo.

**Mini-WARS is a NRP1 ligand that reduces VE-cadherin turnover and impairs histamine-elicited endothelial permeability.** During evolution, aaRSs have acquired key nontranslational functions exerted outside the cell as secreted cytokines[28,29]. These further biological activities of aaRSs arose from the incorporation of new amino acid sequences that are under the inhibitory control of co-evolved domains, which can be removed by alternative splicing or proteolysis[28,29]. Notably, our MS analyses revealed that endothelial NRP1 associates with different aaRSs (Fig. 2c). Among them, WARS is of particular interest because, differently from the native full length (FL) 53 kDa enzyme, both the alternatively spliced 49 kDa mini-WARS fragment and the natural proteolytic 44 kDa T2-WARS fragment, respectively lacking most of or the whole N-terminal WHEP domain, display effective anti-angiogenic function[28,29].

From C- to N-terminus, human FL-WARS displays an anticodon recognition domain, a Rossmann fold catalytic domain shared by WARSs of all species, an eukaryote-specific extension (ESE) shared by all eukaryotic WARSs, and a WHEP domain, which appeared in vertebrate WARS and is connected to ESE by a 21 amino acid long disordered linker[59] (Fig. 5a). Mini-WARS is characterized by the absence of a large N-terminal portion of the WHEP domain, while T2-WARS is devoid of the WHEP domain, the linker, and a part of the ESE[59] (Fig. 5a).

Extracellular T2-WARS localizes at EC intercellular junctions[60] and its adenylate pocket mediates an inhibitory interaction with two N-terminal Trp side chains of VE-cadherin[61] that is sterically interfered by the presence of the WHEP domain[28]. Interestingly, CMT2D patients display point mutations in GARS, which make the full-length synthetase, in presence of a metazoan WHEP domain, capable of binding the b1 domain of NRP1, thus outcompeting with ligands, such as VEGF-A[26,27]. Therefore, we investigated which WARS isoform may bind NRP1 in ECs. Lysates from unstimulated ECs were immunoprecipitated with anti-NRP1 Ab or control non-immune IgGs and analyzed by western blot with an anti-WARS Ab raised against a recombinant fragment containing both the catalytic Rossmann fold and the anticodon recognition domains. Of note, while in cultured ECs FL-WARS was expressed more abundantly than its two shorter anti-angiogenic isoforms, NRP1 robustly and selectively co-immunoprecipitated with mini-WARS only (Fig. 5b). Next, we verified the ability of the purified Fc-tagged recombinant whole extracellular portions of NRP1 to directly interact with recombinant purified tag-free FL, or mini- or T2-WARS (Supplementary Fig. 3c). In vitro pull-down assays on protein G beads clearly showed that the purified Fc-tagged extracellular portion of NRP1 preferentially interacts with purified mini-WARS, minimally with FL and not with T2-WARS (Fig. 5c). Moreover, the simultaneous transduction of cultured ECs with HA-tagged NRP1 and V5-tagged FL-WARS, mini-WARS, or T2-WARS also confirmed that NRP1 associates with V5-mini-WARS, but neither FL-WARS nor T2-WARS (Fig. 5d). To confirm that the cytoplasmic domain was not required for the interaction of NRP1 with mini-WARS, as suggested by our in vitro pull-down analysis (Fig. 5c), lysates of ECs overexpressing HA-tagged wild type (WT) or cytodomain deletion (Δcyto) constructs of NRP1 and V5-mini-WARS were immunoprecipitated with anti-V5 Ab and then blotted with anti-HA Ab (Fig. 5d). We found that the cytoplasmic domain of NRP1 is fully dispensable for its interaction with mini-WARS. Then, we verified whether, similarly to CMT2D GARS mutants[26,27], the interaction with mini-WARS also relies on the extracellular coagulation factor V/VIII homology/b domains of NRP1. We observed that, when overexpressed in ECs, V5-mini-WARS co-immunoprecipitated with HA-tagged WT, but not b1/b2 deletion (Δb1/b2) construct of NRP1 (Fig. 5d). Finally, we verified whether in ECs, similarly to FL WARS[62], the release of cytosolic mini-WARS in the extracellular space may occur via an unconventional pathway involving a multiprotein complex[63] in which the S100A10 protein acts as negative regulator[62]. Indeed, we found that S100A10 silencing in ECs robustly increases the

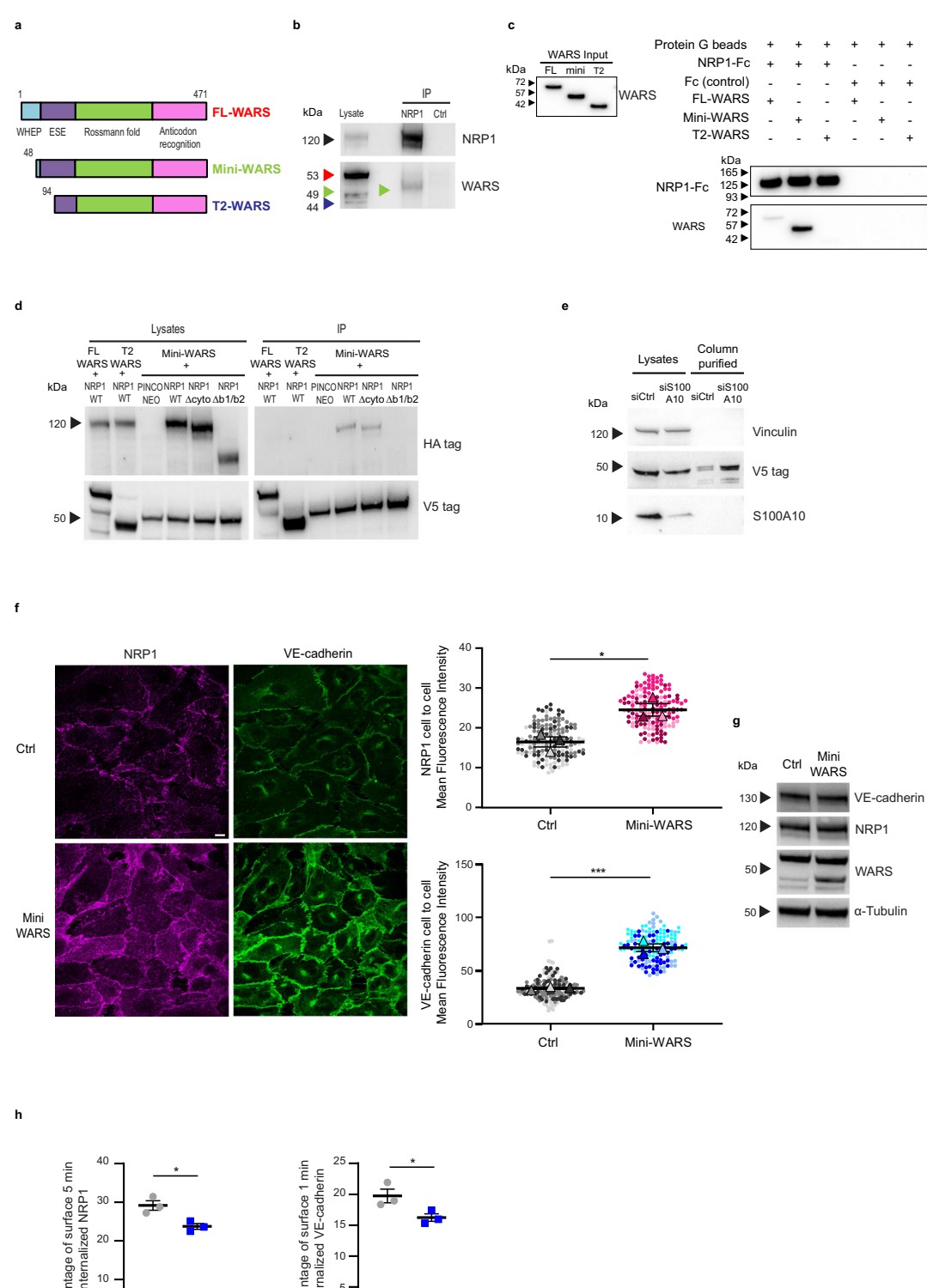

unconventional secretion of transfected V5-mini-WARS in the extracellular medium (Fig. 5e). Therefore, both FL[62] and mini-WARS (this manuscript) rely on an S100A10-regulated unconventional mechanism[63] to be released in the extracellular environment. In sum, similarly to VEGF-A[7,11], SEMA3A[6,7], and CMT2D GARS mutants[26,27], secreted extracellular mini-WARS may also interact with the b1 domain of the extracellular

portion of NRP1 and act as a physiological modulator of NRP1 function.

To understand mini-WARS role in regulating NRP1 function, we overexpressed it in ECs. Interestingly, fluorescent confocal microscopy showed that, higher amounts of NRP1 (1.5-fold ± 0.1%, $P < 0.05$) and VE-cadherin (twofold ± 0.1%, $P < 0.001$) localized at intercellular contacts in mini-WARS overexpressing

**Fig. 5 Mini-WARS is an extracellular inhibitory NRP1 ligand that impairs VE-cadherin turnover. a** Schematic showing the different domains in human Full Length (FL, red), mini (green) and T2 (blue) WARS isoforms; WHEP, vertebrate-specific extension; ESE, eukaryote-specific extension. **b** Endogenous NRP1 and mini-WARS co-immunoprecipitation. An anti-WARS Ab recognizing the catalytic Rossmann fold and the anticodon recognition domain detected only the mini-WARS band (green arrow) co-immunoprecipitating with NRP1. A representative experiment out of five is shown. **c** Pull-down experiment identifies mini-WARS as a preferential binding partner of NRP1-Fc. A representative western blot from three independent experiments is shown. **d** Mini-WARS interaction with NRP1 does not rely on its cytoplasmic domain. A HA antibody used to detect WT or Δcyto or Δb1/b2 NRP1-HA co-immunoprecipitated with FL or T2 or mini-WARS-V5 co-transfected in ECs. The image is representative of three independent experiments. **e** S100A10 silencing increases the secretion of V5-mini-WARS in the EC extracellular medium. Control (siCtrl) or S100A10 (siS100A10) silenced ECs were transfected with mini-WARS V5 tagged and their medium loaded on V5 beads. A representative western blot from three independent experiments is shown. **f** Mini-WARS fosters NRP1 and VE-cadherin localization at endothelial cell-to-cell contacts. Confocal microscopy on pCCL empty lentivirus (Ctrl) or pCCL mini-WARS transduced ECs stained for NRP1 (magenta) and VE-cadherin (green); quantification of NRP1 (above) and VE-cadherin (below) fluorescence intensity in AJs. Data are the mean ± SEM of three independent experiments (depicted in different color shades). Scale bar is 10 μm. Statistical analysis: two-tailed heteroscedastic Student's *t* test; *P value = 0.0164, ***P value = 0.0005. **g** Mini-WARS overexpression does not affect NRP1 and VE-cadherin protein levels in ECs. A representative experiment out of three is shown. **h** Mini-WARS overexpression decreases NRP1 and VE-cadherin internalization in ECs. Percentage of NRP1 (left) and VE-cadherin (right) endocytosis in Ctrl or mini-WARS-transduced ECs as measured by surface labeling followed by capture ELISA. Data are the mean ± SEM of three independent experiments. Statistical analysis: two-tailed heteroscedastic Student's *t* test; *P value = 0.0201 (left), *P value = 0.0498 (right). Source data are provided as a Source Data file.

ECs, compared to control ECs (Fig. 5f). Conversely, the total amounts of both receptor proteins were unaffected by exogenous mini-WARS transduction in ECs (Fig. 5g). Altogether, these data indicate that, in ECs, mini-WARS may bind and stabilize NRP1 at AJs, and that this interaction may hinder VE-cadherin turnover. Supporting this hypothesis, mini-WARS overexpression impaired the endocytosis of both NRP1 (by $19 \pm 0.03\%$, $P < 0.05$) and VE-cadherin (by $18 \pm 0.03\%$, $P < 0.05$) from the surface of the ECs (Fig. 5h). This result suggests that mini-WARS may affect EC intercellular junctions by binding and inhibiting the endocytosis of NRP1 and that this, in turn, may impair VE-cadherin endocytosis and turnover at AJs (Fig. 4a–d). Since we showed that NRP1 promotes the VEGF-A-independent disruption of the endothelial barrier (Fig. 4e) and that mini-WARS avidly binds NRP1 (Fig. 5b–d), we next evaluated if and how histamine-stimulated endothelial permeability was affected by mini-WARS overexpression or silencing of endogenous WARS. We found that histamine-elicited transendothelial flux was diminished when mini-WARS was overexpressed in ECs (Fig. 6a), while it increased in siWARS ECs compared with siCtrl ECs (Fig. 6b). Moreover, the transduction of a silencing-resistant mini-WARS rescued the aberrant permeability of siWARS ECs (Fig. 6b). Finally, we assessed the impact of NRP1 silencing on the ability of overexpressed mini-WARS to impair histamine-stimulated permeability in ECs. We observed that, compared to siCtrl ECs, the overexpression of mini-WARS did not longer hinder the permeability triggered by histamine in siNRP1 ECs (Fig. 6c). Hence, mini-WARS requires NRP1 to exert its inhibitory effect on endothelial permeability, while the presence of VE-cadherin alone was not sufficient in this regard. In sum, mini-WARS represents a physiological NRP1 inhibitory ligand that impairs the internalization of NRP1 and its interactor VE-cadherin, promotes their localization at AJs, and counteracts histamine-elicited endothelial permeability.

## Discussion

NRP1 is crucial for embryonic vascular morphogenesis[3,15], yet its function does not appear to depend on its major ligands SEMA3A[16,17] and VEGF-A[18], which have both shown to exploit NRP1 as co-receptor in ECs. Therefore, NRP1 control of blood vessels may involve different mechanisms and additional extracellular ligands. For instance, NRP1 also acts as an endothelial endocytic receptor that promotes integrin-dependent adhesion dynamics[20,33] and vascular permeability[22].

To unbiasedly define candidate NRP1 regulators, we quantitatively analyzed by MS the network of proteins that associate with NRP1 on the surface and in endosomes of unstimulated

ECs. We discovered that, other than interacting with integrins at ECM adhesion sites[20,36,38], NRP1 also localizes at cell-to-cell contacts[42], where it physically interacts with cell adhesion molecules such as VE-cadherin and PECAM1. Altogether our MS analysis supports the hypothesis that EC adhesion receptors represent a major class of plasma membrane receptors preferentially interacting with NRP1. To organize into a functional blood vascular tree, ECs rearrange their reciprocal positions and compete with each other to reach given strategic locations, such as the tip of angiogenic sprouts[64] in which NRP1 plays a key, but yet poorly understood, role[65]. In this context, the ability of ECs to dynamically remodel cell-to-ECM[8,16,33,66] and cell-to-cell interactions[67] is crucial and relies on the regulation of adhesion receptor conformation and traffic. We have previously shown that NRP1 promotes the internalization of conformationally active integrins at EC-to-ECM contacts[20,33,45]. Here, we provide evidence that NRP1 also bolsters VE-cadherin endocytosis and turnover at intercellular adhesion sites (Supplementary Fig. 4). This function is crucial because it is required for cell rearrangements during angiogenesis[67]. Moreover, it has been shown that it relies on glycolysis[68]. Accordingly, we found that NRP1 may associate with glycolytic enzymes that, in neurons, have been shown to localize on endosomes to facilitate their transport along microtubules[69]. Altogether, our NRP1 interactome makes tempting to speculate that the advent of NRP1 in vertebrate may have allowed ECs and neurons to increase their basal endocytic rate as well as to influence the trafficking dynamics and fate of adhesion receptors, such as integrins and cadherins, to support the development of the cardiovascular apparatus[70] and a more complex nervous system[71–73].

We have observed that in ECs NRP1 promotes the endocytosis of interacting adhesion receptors, e.g. active α5β1 integrins[20] and VE-cadherin (this manuscript), in basal conditions. Moreover, both in vitro and in vivo the lack of NRP1 in ECs impairs vascular permeability elicited by histamine that, independently from NRP1, signals via G-protein coupled histamine receptor H1[57]. However, the direct binding of activatory ligands, such as SEMA3A, further increases the ability of NRP1 to promote adhesion receptor endocytosis and to elicit vascular leakage[25]. Similarly, in ECs VEGF-A[19] and CendR peptides[23] trigger NRP1 internalization and vascular permeability. In addition, SEMA3A-elicited neuron growth cone collapse also relies on NRP1-dependent endocytosis[74,75] and endosomal signaling[76]. Hence, both in ECs and neurons, NRP1 appears to behave as an endocytic receptor whose basal activity controls the turnover of associated transmembrane proteins, such as adhesion receptors, and to be increased by activatory ligands. It has been proposed

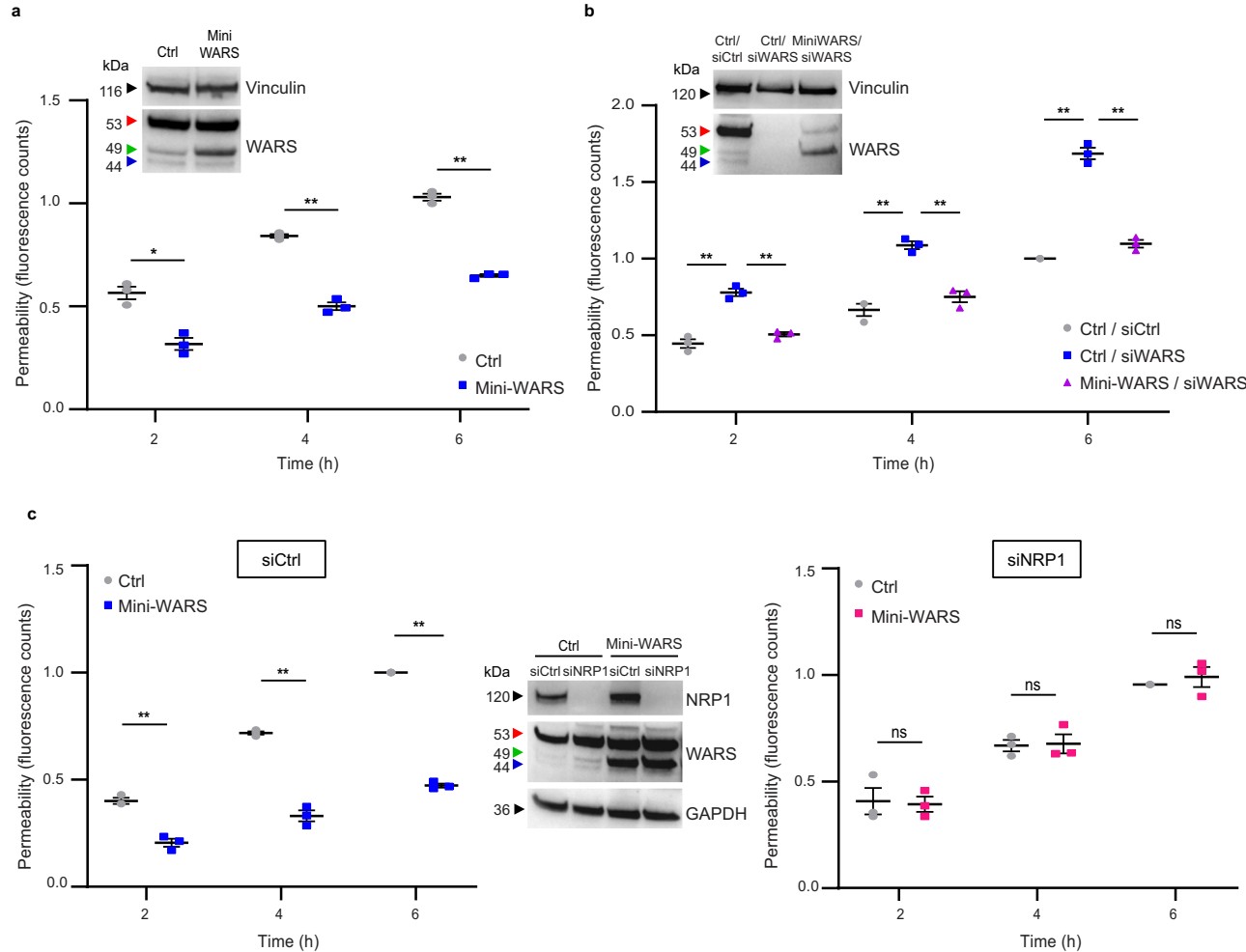

**Fig. 6 Mini-WARS is an extracellular inhibitory NRP1 ligand that impairs histamine-elicited endothelial permeability. a** Mini-WARS overexpression decreases histamine-elicited endothelial permeability. Quantification of 70 kDa Dextran-FITC transendothelial flux over time through a confluent Ctrl (empty lentivirus) or mini-WARS-transduced EC monolayer after histamine stimulation. Data are the mean ± SEM of three independent experiments. Statistical analysis: two-way ANOVA with Bonferroni correction; *$P$ value = 0.0126, **$P$ value = 0.0020, 0.0016. Western blot analysis shows mini-WARS overexpression in ECs. **b** Mini-WARS silencing impairs histamine-elicited endothelial permeability. Quantification of 70 kDa Dextran-FITC transendothelial flux over time through a confluent control (siCtrl) or WARS (siWARS) silenced ECs rescued for empty vector (Ctrl/siCtrl and Ctrl/siWARS) or for mini-WARS (mini-WARS/siWARS) ECs monolayer after histamine stimulation. Data are the mean ± SEM of three independent experiments. Statistical analysis: two-way ANOVA with Bonferroni correction; **$P$ value ≤ 0.01. Western blot analysis shows WARS silencing and mini-WARS overexpression. **c** Mini-WARS overexpression decreases histamine-elicited endothelial permeability via NRP1. Quantification of 70 kDa Dextran-FITC transendothelial flux over time through a confluent control (siCtrl, left) or NRP1 silenced (siNRP1, right) and transduced with empty lentivirus (Ctrl) or mini-WARS EC monolayer after histamine stimulation. Data are the mean ± SEM of three independent experiments. Statistical analysis: two-way ANOVA with Bonferroni correction; ns = $P$ value ≥ 0.05, **$P$ value = 0.0047, 0.0070, 0.0010. Western blot analysis shows NRP1 silencing and mini-WARS overexpression in ECs. Source data are provided as a Source Data file.

that NRP1-dependent endocytosis resembles macropinocytosis[23]. Since phagocytosis and macropinocytosis display several mechanistic similarities[77], it will be worth investigating if NRP1-elicited endocytosis of adhesion receptors impinges on the activation of key phagosomal signaling components that we have identified in our MS analysis as potential NRP1 interactors, e.g., the Arp2/3 protein machinery that drives actin polymerization, or Ezrin and Moesin adaptors that mediate receptor binding to F-actin[77].

Remarkably, we found that in unstimulated ECs NRP1 interacted with at least three different aaRSs, namely GARS, HARS, and WARS, all of which have been causatively linked to CMT disease[27]. A striking feature of these three aaRSs is their shared N-terminal WHEP domain. In fact, these three aaRSs are the only ones with a WHEP domain at the N-terminus[27]. Our previous

structural analysis indicated that the WHEP domain regulates the conformation of GARS and that deletion of the WHEP domain induces a conformational change in GARS that mimics CMT-causing mutants which have gained a capacity to interact with NRP1[26,78]. Consistently, we show here that in ECs mini-WARS but not the FL-WARS with a WHEP domain can interact with NRP1, supporting the concept that the WHEP domain may act as a negative regulator for aaRS-NRP1 interaction. The fact that all three NRP1-interacting aaRSs share a WHEP domain suggest that their interaction with NRP1 may be regulated by the inclusion or exclusion of the WHEP domain.

In addition to the ancient function of ligating an amino acid to the 3'-end of its cognate tRNA, aaRSs acquired during evolution additional translation-independent functions[79] often combined to their unconventional secretion[63] in the extracellular

environment[29]. Of note, both HARS[80] and WARS[30,81,82] were characterized for their anti-angiogenic properties. In the case of FL-WARS, removal of the N-terminal WHEP domain-containing portion, either by alternative splicing or proteolytic cleavage, gives rise to mini-WARS and T2-WARS respectively that behave as potent secreted anti-angiogenic factors[30,81,82]. In particular, T2-WARS anti-angiogenic function relies on its tryptophan and adenosine pocket-mediated binding to Trp2 and Trp4 of VE-cadherin N-terminal extracellular domain[60,61]. Here, we provide evidence that in ECs mini-WARS, but neither FL nor T2-WARS, effectively binds the extracellular segment of NRP1 (Supplementary Fig. 4). Moreover, our finding that mini-WARS impairs NRP1 endocytosis and stabilizes its localization at AJs implies that mini-WARS functions as an inhibitory ligand of NRP1 (Supplementary Fig. 4). Furthermore, mini-WARS over-expression in ECs requires the presence of NRP1 to inhibit the histamine-elicited endothelial permeability, while VE-cadherin alone was not sufficient in this regard (Supplementary Fig. 4).

Here, we reveal that, similarly to CMT2D GARS mutants[26,27], VEGF-A[7,11], and SEMA3A[6,7], mini-WARS also binds to the extracellular coagulation factor V/VIII homology/b domains of NRP1. Our previous[20] and present data suggest that in ECs the internalization-dependent turnover of mechanosensing adhesion receptors, such as integrins[20] and VE-cadherin (this manuscript), may be under the control of NRP1. In turn, NRP1 pro-endocytic function may be regulated by activating (VEGF-A and SEMA3A) or inhibiting (mini-WARS and CMTD2 GARS mutants) ligands (Supplementary Fig. 4). Our observation that mini-WARS impairs NRP1 function in the absence of exogenously added VEGF-A implies that mini-WARS directly inhibits NRP1 through yet unknown mechanisms (Supplementary Fig. 4), such as the allosteric modulation of NRP1 extracellular conformation and/or interaction with other transmembrane proteins. Further work is required to thoroughly characterize the role of NRP1-interacting aaRSs.

Abnormal VE-cadherin-mediated endothelial intercellular contacts[83] make cancer blood vessels hyperpermeable, triggering increased interstitial fluid pressure, reduced anti-cancer drug delivery, and enhanced metastatic spread[9,84]. Therefore, the ability of mini-WARS to stabilize VE-cadherin at AJs and to impair vascular permeability may be therapeutically exploited to normalize the cancer vasculature[9,85].

## Methods

**DNA constructs**. PINCO-Nrp1 WT and PINCO-Nrp1 Δcyto constructs were previously described[20]. The PINCO-Nrp1 Δb1/b2 construct was generated by deleting in Nrp1 WT the b1 and b2 domains using the Phusion™ high-fidelity DNA polymerase (ThermoFisher Scientific) and the following oligonucleotide primers: 5'-CTGAAGATTTTAAGTGTGCTGGACCAACCACAC-3' (forward); 5'- GTG TGGTTGGTCCAGCACACTTAAAATCTTCAG-3' (reverse). The HaloTag encoding sequence (derived from Promega) was fused immediately 3' to that of the signal peptide of PINCO-NRP1 WT, by standard PCR protocols. FL-WARS, mini-WARS, and T2-WARS constructs were cloned into pcDNA6/V5-His C vector (ThermoFisher Scientific, catalog # V22020). HaloTag-NRP1 and mini-WARS were subcloned into a third-generation lentiviral vector pCCL.sin.cPPT.PGK.GFP.WPRE[86,87] (pCCL), with In-Fusion 2.0-CF Dry-DownPCR Cloning-Kit (catalog no. 639607; Clontech Laboratories, Mountain View, CA). VE-cadherin-mCherry in the lentiviral vector HIV3 was kindly donated by Elisabetta Dejana (IFOM, Milano, Italy).

**Cell culture**. Primary human ECs were isolated from the umbilical cords. Briefly, umbilical arteries were cannulated with a blunt 17-gauge needle that was secured by clamping the cord over the needle. The umbilical arteries were then perfused with 50 ml of phosphate-buffered saline (PBS) to wash out the blood. Next, 10 ml of 0.2% collagenase A (Cat. # 11088793001, Roche Diagnostics) diluted in cell culture medium were infused into the umbilical arteries and incubated 30 min at room temperature. The collagenase solution containing the ECs was flushed from the cord by perfusion with 40 ml of PBS, collected in a sterile 50 ml centrifuge tube, and centrifuged 5 min at 800 × g. Cells were first resuspended in Endothelial Cell Growth Basal Medium (EBM-2) supplemented with EGM-2 BulletKit (Lonza) (EGM-2), and subsequently plated in cell culture dishes that had been previously

adsorbed with 1% gelatin from porcine skin (G9136, Sigma-Aldrich). Cells were tested for mycoplasma contamination by means of Venor GeM Mycoplasma Detection Kit (MP0025-1KT, Sigma-Aldrich) and grown in EGM-2 medium. The isolation of primary arterial ECs from human umbilical cords was approved by the Office of the General Director and Ethics Committee of the Azienda Sanitaria Ospedaliera Ordine Mauriziano di Torino hospital (protocol approval no. 586, October 22, 2012 and no. 26884, August 28, 2014) and informed consent was obtained from each patient.

**Antibodies and reagents**. Mouse monoclonal antibody (mAb) anti-HaloTag (Promega, G921A, for western blot—WB- 1:1000); mAb anti-α tubulin (Sigma-Aldrich, T5168/clone B-5-1-2, WB 1:8000); goat polyclonal anti-EEA1 (Santa Cruz Biotechnology, N-19/sc-6415, immunofluorescence-IF-1:200); rabbit anti VEGFR-2 (Cell Signaling, 55B11/#2479, WB 1:1000); mAb anti-NRP1 (R&D Systems, MAB3870, IF 1:50, live for immunoprecipitation-IP-1:2000); goat polyclonal anti-VE-cadherin (Santa Cruz Biotechnology, C-19/sc-6458, IF 1:200, ELISA 1 μg/ml); rabbit anti-VE-cadherin (Cell Signaling, D87F2, WB 1:1000); rabbit anti-NRP1 (Abcam, EPR3113, WB 1:3000); mAb anti-PECAM1 (Cell Signaling, 89C2/#3528, WB 1:1000, IF 1:50); goat polyclonal anti-NRP1 (Santa Cruz Biotechnology, C19/sc-7239, IP 1 μg/ml lysate); rabbit polyclonal anti-WARS (ThermoFisher Scientific, PA5-29102, WB 1:1000); mAb anti-vinculin (Sigma-Aldrich, V9131, WB 1:2000); rabbit anti-V5 (Cell Signaling, D3H8Q, IP 1 μg/ml lysate); rat monoclonal Ab anti-HA (Roche, 11867423001, WB 1:1000); mAb anti-V5 (Santa Cruz Biotechnology, E10/sc-81594, WB 1:1000); mAb anti S100A10 (Santa Cruz Biotechnology, 4E7E10/sc-81153, WB 1:500); mAb anti-transferrin receptor 1 (Santa Cruz Bio-technology, CD71/G-8/sc-393719, ELISA 1 μg/ml); polyclonal sheep anti-NRP1 (R&D Systems, AF3870, ELISA 1 μg/ml); rabbit monoclonal Ab anti-Nrp1 (Abcam, ab81321, mouse tissue IF 1:100); rat monoclonal Ab anti-panendothelial cell antigen Meca32 (BD Pharmingen, 550563, mouse tissue IF 1:100); goat anti-rabbit secondary Ab (Jackson ImmunoResearch Laboratories, 111-035-003, WB 1:20000); goat anti-mouse secondary Ab (Jackson ImmunoResearch Laboratories, 115-035-003, WB 1:20000); goat anti-rat HRP (Abcam, ab97057, WB 1:2000); Alexa Fluor-488 donkey anti-mouse (A21202) and anti-goat (A11055) IgG (H + L) secondary Ab (Life Technologies, IF 1:400); Alexa Fluor-555 donkey anti-mouse (A31570) and anti-goat (A21432) IgG (H + L) secondary Ab (Life Technologies, IF 1:400); Alexa Fluor-647 donkey anti-mouse (A31571) and anti-goat (A21447) IgG (H + L) secondary Ab (Life Technologies, IF 1:400); DAPI (Life Technologies, D3571); normal Goat (NI02-100UG) and mouse (NI03-100UG) IgGs (Millipore, IP 1 μg/ml lysate); mouse monoclonal agarose-conjugated anti-NRP1 antibody (Santa Cruz Biotechnology, A-12/sc-5307, 1 μg pull-down assay); goat anti-human IgG Fc HRP preadsorbed (Abcam, ab98624, WB 1:10000); agarose anti-V5 tag antibody (Abcam, ab1229, 0.2 μg/ml medium); mAb anti-GAPDH (Abcam, 6C5/ab8245, WB 1:10000). HaloTag Alexa 660-labeled fluorescent chloroalkane Ligand (Promega, G8471, 1:200); HaloTag PEG-Biotin Ligand for surface labeling (Promega, G8591, 2.5 μM); Streptavidin-Agarose conjugate beads (Millipore, 16-126); Histamine (Sigma-Aldrich, H7125, 100 μM); Fluorescein isothiocyanate (FITC)-Dextran 70 (Sigma-Aldrich, 46945, 25 mg/ml); 0.4 μm pore size costar transwell (Sigma-Aldrich, 3470); Dynabeads™ protein G beads (ThermoFisher Scientific, 10003D, 2 μl/0.1 μg recombinant protein); Amicon Ultra-30k Centrifugal Filters (Millipore, Z717185).

**Recombinant proteins**. Human plasma fibronectin (R&D Systems, 1918-FN-02M, 3 μg/ml), human NRP1-Fc (R&D Systems, 10455-NI-050), human VE-Cadherin-Fc (R&D Systems, 938-VC-050) and human IgG1-Fc (R&D Systems, 110-HG-100); recombinant human VEGF-A165 (R&D Systems, 293-VE).

For the expression and purification of recombinant human WARS isoforms, DNA encoding human FL-WARS, mini-WARS, or T2-WARS with a N-terminal His6-SUMO tag was cloned into pET-28a (+) vector (Novagen). WARS expression was induced in E. coli BL21(DE3) cells with 1 mM isopropyl beta-D-thiogalactopyranoside. Following cell lysis, WARS was purified using Ni-NTA beads (Qiagen). The N-terminal His6-SUMO tag was cleaved with sumo protease ULP1. The resulting tag-free WARS proteins were further purified using HiLoad 16/60 Superdex 200 prep grade column (GE Healthcare).

**Gene silencing**. For siRNA-mediated silencing, the day before oligofection, ECs were seeded in six-well dishes at a concentration of $10 \times 10^3$ cells/well. Oligofection of siRNA duplexes was performed according to the manufacturer's protocols. Briefly, human ECs were transfected twice (at 0 and 24 h) with 200 pmol of siGENOME Non-Targeting siRNA Pool #1 (D-001206-13) as control (siCtrl) or a mix of three (in the case of human NRP1 and WARS), or a mix of four (for human S100A10) siRNA oligonucleotides (GE Healthcare Dharmacon), using Oligo-fectamine Transfection Reagent (ThermoFisher Scientific). Twenty-four hours (NRP1), 48 h (WARS), or 96 h (S100A10) after the second oligofection, ECs were lysed or tested in functional assays. Knockdown of human S100A10 was achieved through the single oligonucleotide sequences: (1) 5'-GAUAAAGGCUACUUAAC AAUU-3'; (2) 5'-GAAAAGGAGUUCCCUGGAUUU-3'; (3) 5'-GAACACGCCA UGGAAACCAUU-3'; (4) 5'-GGACCAGUGUAGAGAUGGCUU-3'. While, in the case of human WARS, the single oligonucleotide sequences were: (1) 5'-GGUUCC AAGUAUACUCUUAUU-3'; (2) 5'-CCAAAUGACCCUAGACCCUUU-3'; (3) 5'-CUGGAAGAGCAAAGCCAAAUU-3'; in the case of human NRP1, the single

oligonucleotide sequences were: (1) 5′-AAUCAGAGUUUCCAACAUA-3′; (2) 5′-GAAGGAAGGGCGUGUCUUG-3′; (3) 5′-GUGGAUGACAUUAGUAUUA-3′.

**Live cells labeling of HaloTag-NRP1 using fluorescent HaloTag Alexa 660 ligand and confocal microscopy.** The day before ECs transduced with pCCL HaloTag-NRP1 were plated on glass coverslips coated with gelatin 1%. Cells were incubated for 15 min at 4 °C (to inhibit endocytosis) with the cell impermeable HaloTag Alexa 660 fluorescent ligand (dilution 1:200), and then shifted or not to 37 °C for 3 min. ECs were washed in warm fresh medium twice, fixed in 1% paraformaldehyde (PFA), permeabilized in 0.01% saponin for 5 min on ice, and were stained with goat anti-EEA1 (dilution 1:200) and Alexa Fluor secondary Ab (dilution 1:400). Cells were analyzed by using a Leica TCS SP2 AOBS confocal laser-scanning microscope (Leica Microsystems).

**Live cells labeling of HaloTag-NRP1 using HaloTag PEG-Biotin ligand and immunoprecipitation.** Transduced ECs were starved for 3 h with EBM-2 medium (Lonza) at 37 °C, 5% $CO_2$ in a humidified atmosphere. Cells were surface labeled with the cell impermeable HaloTag PEG-Biotin Ligand 3 µM at 4 °C for 15 min, stimulated or not with VEGF-A165 30 ng/ml for 10 min at 37 °C. ECs were lysed in buffer containing 25 mM Tris–HCl pH 7.6, 100 mM NaCl, 1% Triton X-100, 5% glycerol, 0.5 mM EGTA, 2 mM $MgCl_2$, 1 mM PMSF, 1 mM $Na_3VO_4$ and protease inhibitor cocktail. Cellular lysates were incubated for 20 min on wet ice, and then centrifuged at 15,000 × $g$, 20 min, at 4 °C. The total protein amount was determined using the bicinchoninic acid (BCA) protein assay reagent (Pierce). Equivalent amounts (1 mg) of protein were immunoprecipitated for 2 h at 4 °C on streptavidin-conjugated beads. Immunoprecipitates were washed three times with lysis buffer with or without detergent and then separated by SDS–PAGE with Mini PROTEAN TGX precast 7.5% gel (BIO-RAD). Proteins were then transferred to a Trans-Blot Turbo™ PVDF Transfer (BIO-RAD), probed with antibodies of interest and detected by enhanced chemiluminescence technique (PerkinElmer).

**Adhesion assay.** In all, 6000 transduced and silenced ECs were resuspended in 0.1 ml of EBM-2 medium (Lonza) and plated on 96-well microtiter plates (Costar) that were previously coated with 3 µg/ml FN and then saturated with 3% bovine serum albumin (BSA). Cells were left to adhere for 20 min. Cells were fixed in 8% glutaraldehyde and then stained with 0.1% crystal violet, 20% methanol. Cells were photographed with a QIcam FAST1394 digital color camera (QImaging) and counted by means of Image-ProPlus 6.2 software (Media Cybernetics).

**Pull-down assay for MS analysis.** ECs transduced with pCCL lentivirus carrying HaloTag-NRP1 construct were surface labeled or not (Ctrl) with the cell impermeable HaloTag PEG-Biotin Ligand 2.5 µM at 4 °C for 15 min and shifted or not at 37 °C for 3 min. Cells were lysed in buffer containing 25 mM Tris–HCl pH 7.6, 100 mM NaCl, 1% Triton X-100, 5% glycerol, 0.5 mM EGTA, 2 mM $MgCl_2$, 1 mM PMSF, 1 mM $Na_3VO_4$ and protease inhibitor cocktail. Cellular lysates were incubated for 20 min on wet ice, and then centrifuged at 15000x$g$, 20 min, at 4 °C. The total protein amount was determined using the bicinchoninic acid (BCA) protein assay reagent (Pierce). Equivalent amounts (3 mg) of protein were incubated for 2 h at 4 °C on streptavidin-conjugated beads. Proteins captured on streptavidin-conjugated beads were then washed three times with lysis buffer with and without detergent and eluted with reducing sample buffer.

**MS analysis.** Proteins eluted in sample buffer were boiled 5 min at 95 °C and separated on 4–12% gradient NuPAGE Novex Bis-Tris gel (Life Technologies) and in-gel digested[88] with trypsin (Promega). Digested peptides were desalted using StageTip[89], and injected on line to a LTQ-Orbitrap Elite via a nanoelectrospray ion source (Thermo Scientific). Peptides were separated using a 20-cm fused silica emitter (New Objective) packed in house with reversed-phase Reprosil Pur Basic 1.9 µm (Dr. Maisch GmbH), and eluted with a flow of 200 nl/min from 5 to 30% of buffer containing 80% ACN in 0.5% acetic acid, in a 90 min linear-gradient, as previously described[90]. The mass range acquired for the full MS scan was 300–1650 $m/z$ with a resolution of 120,000 (HCD mode) at 400 Th and the Orbitrap aimed to collet $1 \times 10^6$ charges at a time. The top ten most intense peaks in the full MS were isolated for fragmentation with a target of 40,000 ions at a resolution of 15,000 at 400 Th. Ions singly charged were excluded, whilst the ions that have been isolated for MS/MS were subsequently added on an exclusion list. MS data were acquired using the Xcalibur software (Thermo Scientific) and .raw files processed with the MaxQuant computational platform[91] version 1.5.0.26 and searched with the Andromeda search engine[92] against the human UniProt database[35] (release-2012 01, 88,847 entries). To search the parent mass and fragment ions we required a mass deviation of 4.5 ppm and 20 ppm. The minimum peptide length was seven amino acids and maximum of two missed cleavages and strict specificity for trypsin cleavage were required. Carbamidomethylation (cysteine) was set as fixed modification, whereas oxidation (methionine) and N-acetylation were set as variable modifications. The false discovery rates at the protein and peptide levels were set to 1%. The requantification and match between runs features were enabled. For protein quantification, label-free quantification (LFQ) was calculated by MaxQuant[34]. Unique and razor

peptides were used for protein quantification and we required proteins to be quantified with at least two ratio counts.

**MS data analysis.** The proteinGroup.txt output of MaxQuant was analyzed with Perseus[93]. Common reverse and contaminant hits (as defined in MaxQuant) were removed, as well as proteins only identified by site. The LFQ Intensity values were used for protein quantification. Only proteins quantified in at least three of the four biological replicates in at least one experimental group were kept for follow-up analysis. LFQ values were log2 scaled and missing values were replaced using the Imputation function (set up: from normal distribution, width 0.3 and downshift 1.8). For each protein, its enrichment in the NRP1 surface interactome and NRP1 endosomal interactome was calculated by dividing the LFQ intensity to the LFQ intensity measured in the control sample (Ctrl = HaloTag-NRP1-transduced ECs that were not surface labeled with HaloTag PEG-Biotin Ligand). A one-sample $t$ test on the calculated ratios (NRP1 surface/Ctrl or NRP1 endosomal/Ctrl) was used to calculate $P$ values and generate the volcano plots. Proteins that had a fold increase of at least 1.5 compared to the control and $P$ value lower than 0.05 were considered potential NRP1 interactors. In Supplementary Data, we also reported the adjusted $P$ values ($q$-values), which were calculated using Benjamini–Hockbert FDR. To exclude that the potential NRP1 interactors were proteins commonly found as background in affinity purification experiments using the HaloTag, we compared the list of NRP1 interactors to those found in the CRAPome of HaloTag affinity purification experiments (www.crapome.org). Potential background proteins have been highlighted in Supplementary Data 1.

**Data analysis.** The selection of transmembrane proteins from the NRP1 interactome was based on the UniProt[35] "Topology" annotation. The "UniProt Annotation Retriever" software tool [https://github.com/gkoulouras/UniProt-annotation-retriever] was used to fetch relevant protein annotation. Custom R scripts were developed, and the following terms were used: "Single-pass type I membrane protein", "Single-pass type II membrane protein" and "Multi-pass membrane protein". Based on their "subcellular locations", as defined by UniProt, the selected transmembrane proteins were grouped into the following groups "Membrane", "Focal adhesion", "Cell membrane", and "Cell Junction". Endosomal proteins were selected based on the subcellular location classification defined by UniProt, in particular we selected the categories "Early endosome", "Late endosome", "Lysosome", and "Cytoplasmic vesicles".

STRING[94] was used for category enrichment analysis (KEGG[95] functional annotation categories, which were reported in the Analysis output of STRING, were used) and to generate the interaction network of the NRP1 interactome with transmembrane and endosomal proteins excluded. For Supplementary Fig. 2b–d, the categories found significantly enriched (FDR < 0.05) were grouped in three main categories as indicated in Supplementary Data 2. The physical and functional interactions of the networks generated with STRING were defined based on default parameters, with the minimum required confidence score of 0.400, but using only "Neighborhood", "Experiments", and "Databases" as active interaction sources.

The chord diagrams and networks in Fig. 2b and Supplementary Fig. 2d were generated with R using the package "circlize[96]" (source code: https://github.com/jokergoo/circlize) and "igraph[97]" (https://igraph.org/r/), respectively.

**Confocal microscopy.** ECs were plated on glass coverslips coated with 1% gelatin from porcine skin (G9136, Sigma-Aldrich) and allowed to adhere overnight or plated on fibronectin (3 µg/ml) to reach the required confluence to observe cell-to-cell contacts. Cells were washed in calcium ($Ca^{2+}$) and magnesium ($Mg^{2+}$) supplemented PBS, fixed in 2% paraformaldehyde (PFA), permeabilized in 0.1% Triton X-100 for 2 min on ice or fixed in Methanol at −20 °C for 10 min. Cells were incubated with different primary Abs for 1 h and revealed by appropriate Alexa Fluor- secondary Ab. Cells were analyzed by using a Leica TCS SPE confocal laser-scanning microscope (Leica Microsystems). We quantified the Mean Fluorescence Intensity in AJs drawing linear regions of interest (ROI) by means of the Leica Confocal Software Quantification Tool (Leica LAS-X). Image acquisition was performed by adopting a laser power, gain, and offset settings that allowed maintaining pixel intensities (grayscale) within the 0–255 range and hence avoid saturation.

**Immunoprecipitation and western blot analysis.** ECs were seeded in 150 mm dishes coated with 1% gelatin coating. ECs were lysed in buffer containing 25 mM Tris–HCl pH 7.6, 150 mM NaCl, 1% NP-40, 5% glycerol, 0.5 mM EGTA, 2 mM $MgCl_2$, 1 mM PMSF, 1 mM $Na_3VO_4$ and protease inhibitor cocktail. Cellular lysates were incubated for 20 min on wet ice, and then centrifuged at 15,000 × $g$, 20 min, at 4 °C. The total protein amount was determined using the bicinchoninic acid (BCA) assay (Pierce). Equivalent amounts (1 mg) of protein were immunoprecipitated for 1 h at 4 °C with the antibody of interest, and immune complexes were recovered on protein A- or G-Sepharose (GE Healthcare) for 1 h at 4 °C. To co-immunoprecipitate VE-cadherin and PECAM1 with NRP1, ECs were incubated live with the mouse mAb that recognize the extracellular domain of NRP1 or control mouse IgGs for 10 min at 37 °C and immune complexes were recovered on protein A- or G-Sepharose (GE Healthcare) for 2 h at 4 °C. Immunoprecipitates were washed three times with lysis buffer with or without detergent and then separated by SDS–PAGE. Proteins were then transferred to a Trans-Blot Turbo

Nitrocellulose membrane (BIO-RAD), probed with antibodies of interest, and detected by Clarity Western ECL Substrate (BIO-RAD).

**Pull-down assays**. For evaluating direct NRP1 and VE-Cadherin interaction, 2 µg of recombinant Fc-tagged whole extracellular portion of human NRP1 (NRP1-Fc, R&D Systems) were incubated with 2 µg of recombinant Fc-tagged whole extracellular portion of human VE-Cadherin (VE-cadherin-Fc) or with the human IgG1-Fc alone on agarose-conjugated anti-NRP1 antibody (1 µg) for 2 h at 4 °C, in 25 mM Tris–HCl buffer pH 7.4, 100 mM NaCl, 5 mM CaCl$_2$. For evaluating the direct interaction of NRP1 with WARS proteins, 0.1 µg of the recombinant NRP1-Fc or human IgG1-Fc protein was immobilized on 2 µl of Dynabeads™ protein G beads (ThermoFisher Scientific) for 30 min at 4 °C. Dynabeads coated with NRP1-Fc or Fc control were then incubated with 0.5 µg of recombinant tag-free human FL-, mini-, or T2-WARS for 2 h at 4 °C. Samples were washed three times with wash buffer (50 mM Tris–HCl, 150 mM NaCl, 1% Triton X-100, 5 mM CaCl$_2$). Lastly, samples were eluted with 50 µl of 1×LDS containing 5 mM DTT and analyzed by western blotting.

**Fluorescence recovery after photobleaching (FRAP)**. Transduced and silenced ECs, plated in WillCo Glass Bottom Dishes, 1% gelatin from porcine skin (G9136, Sigma-Aldrich)-coated, were analyzed by using a Leica TCS SP8 AOBS confocal microscope equipped with ×63 (HC PL APO CS2 ×63/1.40 oil) objective, 37 °C humidified chamber with 5% CO$_2$, and hybrid detectors. The 488 nm line of Argon laser at high intensity was used for beaching the selected ROI, and 2 pre-bleach, 2 bleach, and 50 post-bleach frames (every 1.4 s) were acquired.

**Internalization assay**. Cells were transferred to ice, washed twice in cold phosphate-buffered saline (PBS), and surface labeled at 4 °C with 0.2 mg/ml sulfo-NHS-SS-biotin (Pierce) in PBS for 30 min. Labeled cells were incubated in cold PBS and transferred to prewarmed EBM-2 containing 2% FBS at 37 °C. At the indicated times, the medium was removed, and dishes were rapidly transferred to ice and washed twice with ice-cold PBS. Biotin was removed from proteins remaining at the cell surface by incubation with a solution containing 20 mM sodium 2-mercaptoethanesulfonate (MesNa) in 50 mM Tris–HCl (pH 8.6), 100 mM NaCl for 1 h at 4 °C. MesNa was quenched by the addition of 20 mM iodoacetamide (IAA) for 10 min, and after other two further washes in PBS, the cells were lysed in 25 mM Tris–HCl, pH 7.4, 100 mM NaCl, 2 mM MgCl$_2$, 1 mM Na$_3$VO$_4$, 0.5 mM (ethylene glycol tetraacetic acid) EGTA, 1 % Triton X-100, 5% glycerol, protease inhibitor cocktail (Sigma-Aldrich, 50 mg/ml pepstatin, 50 mg/ml leupeptin, and 10 mg/ml aprotinin), and 1 mM phenylmethanesulfonylfluoride (PMSF). Lysates were cleared by centrifugation at 12,000 × g for 20 min. Levels of biotinylated VE-cadherin, NRP1 or transferrin receptor 1 were determined by capture ELISA assay.

**Capture ELISA assay**. Corning 96 Well Clear Polystyrene High Bind Stripwell Microplate (product #2592, Corning) were coated overnight with 1 µg/ml of appropriate anti-VE-cadherin, NRP1 or transferrin receptor 1 antibodies in 0.05 M Na$_2$CO$_3$ (pH 9.6) at 4 °C and were blocked in PBS containing 0.01% Tween-20 (PBS-T) with 5% BSA for 1 h at RT. Proteins were captured by overnight incubation of 50 µl cell lysate at 4 °C. Unbound material was removed by extensive washing with PBS-T, and wells were incubated with streptavidin-conjugated horseradish peroxidase (GE Healthcare) in PBS-T containing 1% BSA for 1 h at 4 °C. Following further washing, biotinylated receptors were detected by a chromogenic reaction with ortho-phenylenediamine.

**In vitro permeability assay**. ECs were seeded on 6.5-mm diameter Cell Culture Inserts contain PET membrane with 0.4-µm pore size (Costar) pre-coated with 1% gelatin or fibronectin (3 µg/ml), cultured for 24 h in complete culture medium, starved in EBM 2 % FBS 1% BSA for 3 h and assayed for permeability to fluorescein isothiocyanate (FITC)-dextran (70 kDa) (Sigma-Aldrich). When cells were confluent, 25 mg/ml of FITC-dextran with the stimulus Histamine 100 µM, were added to the medium (EBM 2% FBS) of the insert apical compartment. At 2 h, 4 h, and 6 h, 50 µl aliquots of the medium were collected from the basal compartment and the paracellular flux was measured as the amount of FITC-dextran in the medium using the fluorometer Synergy HTX Multi-Mode Microplate Reader (BioTek Instruments).

**In vivo vascular permeability assay**. All animal procedures were approved by the ethics committee of the University of Torino and by the Italian Ministry of Health (Protocol approval no. 337/2016-PR) and also by the Institutional Animal Care and Research Advisory Committee of the KU Leuven. Endothelial-specific Nrp1 knock-out mice (Nrp1$^{EC-/-}$) were generated in a C57BL/6 background by crossing floxed conditional Nrp1-null (Nrp1$^{fl/fl}$) mice [B6.129 (SJL) Nrp1tm2Ddg/J[12] from The Jackson Laboratories] with Cdh5(PAC)-CreERT2[98] mice that carry a tamoxifen-inducible Cre deleter under the control of the VE-cadherin promoter Cdh5(PAC) and were provided from Cancer Research Technology Ltd, according with the Agreement procedures. Gene deletion in adult mice was induced as described previously[99]. In order to analyze junction integrity after treatment, 10 µg of histamine dissolved in 50 µl PBS was injected subcutaneously in the right flank of Nrp1$^{EC-/-}$ and control Nrp1$^{+/+}$ mice, while PBS injected in the left flank of the same animal served as a negative control. 10 min later, 3 mg of Evans Blue

dissolved in 100 µl PBS per 10 g of body weight was injected intravenously. Following 20 min incubation, residual Evans Blue within blood vessels was removed by heart perfusion using saline. Equal-sized skin pieces covering the injected areas were resected using a skin biopsy punching tool and digested in 400 µl formamide for 48 h at 60 °C. Afterward, the amount of extravasated Evans Blue was quantified using a spectrophotometer at a wavelength of 620 nm. Tissue samples were then dried for 96 h at 60 °C, weighted and the amount of extravasated Evans Blue was normalized to (a) its dry weight and (b) the ratio of the sample's weight and the lightest sample from the same animal.

Endothelial-specific Nrp1 knockdown was evaluated by immunofluorescence and confocal microscopy analysis. To this aim, ear tissues were fresh frozen in OCT. Cryostat sections (10-µm) were air-dried, fixed in zinc-fixative (6.05 g Tris, 0.35 g Ca(C$_2$H$_3$O$_2$)$_2$, 2.5 g Zn(C$_2$H$_3$O$_2$)$_2$, 2.5 g ZnCl, 3.8 ml HCl 37%) for 10 min and blocked in 3% BSA and 5% donkey serum in 1×PBS. Tissue sections were incubated with rabbit monoclonal Ab anti-Nrp1 (ab81321) from Abcam and rat monoclonal Ab anti-panendothelial cell antigen Meca32 (550563) from BD Pharmingen, both diluted 1:100 in saturation solution. Anti-rabbit Alexa Fluor-555 and anti-rat Alexa Fluor-488 were employed as secondary Abs. Nuclei were counterstained with DAPI (Invitrogen). All immunofluorescence images were captured by using a TCS SPE confocal laser-scanning microscope (Leica Microsystems), and by maintaining the same laser power, gain, and offset settings.

**Mini-WARS secretion**. ECs were seeded in six-well dishes (12 wells for condition) and transfected twice with siCtrl or siS100A10 oligonucleotides. 72 h after the second oligofection, ECs were lipofectamine transfected with 1 µg/ml of V5-tagged mini-WARS. After 24 h, culture supernatants were collected, concentrated using Amicon Ultra-30k (Millipore), immunoprecipitated on 10 µl of Agarose Anti-V5 tag antibody (Abcam ab1229) for 2 h at 4 °C and analyzed by western blot analysis.

**Statistics**. For statistical evaluation, data distribution was assumed to be normal. Parametric two-tailed heteroscedastic Student's $t$ test was used to assess the statistical significance when two groups of unpaired normally distributed values were compared; when more than two groups were compared, parametric two-tailed analysis of variance (ANOVA) with Bonferroni correction was applied. For each experiment, we performed three independent biological replicates, in which at least three technical replicates were performed. Then, to analyze the reproducibility, we calculated the mean of the three independent biological replicates ± SEM and performed statistical analyses. Statistical differences were considered not significant (ns) = $P$ value > 0.05; significant *$P$ value ≤ 0.05; **$P$ value ≤ 0.01; ***$P$ value ≤ 0.001).

**Reporting summary**. Further information on research design is available in the Nature Research Reporting Summary linked to this article.

## Data availability

The authors declare that all data, including uncropped blots, supporting the findings of this study are available within the article and its Source Data files. The Source Data for Supplementary Fig. 2 can be found in Supplementary Data 3. The .raw MS files and search/identification files generated in this study with MaxQuant computational platform[91] version 1.5.0.26 have been deposited to the ProteomeXchange Consortium via the PRIDE partner repository[100] under the dataset identifier PXD019700. STRING was used for category enrichment analysis (KEGG functional annotation categories, which were reported in the Analysis output of STRING, were used). Source data are provided with this paper.

## Code availability

The source code and the executable that have been used to annotate the topology and subcellular location of NRP1 interactors can be found in the following GitHub repository (https://doi.org/10.5281/zenodo.6645639).

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

## Acknowledgements

The research leading to these results has received funding from: AIRC under IG 2018—ID. 21315—P.I. Serini Guido, IG 2017—ID. 20366—P.I. Valdembri Donatella, IG 2017—ID. 19957—P.I. Enrico Giraudo; AIRC under 5 per Mille 2018—ID. 21052 program—P.I. Comoglio Paolo, G.L.s Serini Guido and Enrico Giraudo (to G.S. and E.G.); FPRC-ONLUS Grant "MIUR 2010 Vaschetto—5 per mille 2010 MIUR" (to G.S.); Telethon Italy (GGP09175) (to G.S.); Associazione 'Augusto per la Vita' (to G.S.); Ministero dell'Istruzione, dell'Università e della Ricerca (PRIN 2020EK82R5) (to G.S.); Università di Torino, Bando Ricerca Locale 2019 (CUP D84I19002940005) (to G.S.); PTCRC-Intra 2020 FPRC 5xmille 2017 Ministero Salute, project "SEE-HER" (to E.G.); CRUK Beatson Institute A31287, CRUK Glasgow Centre A18076 and Stand Up to Cancer campaign for Cancer Research UK A29800 (to S.Z.); long-term structural Methusalem funding from the Flemish government (to M.M.).

## Author contributions

G.S. conceived the project; G.S., S.Z., D.V., and X.L.Y. contributed equally to develop the project; G.S., N.G., N.W., W.C., B.K., M.E., M.M., X.L.Y., D.V., and S.Z. designed the experiments; G.S., X.L.Y., M.M., E.G., D.V., and S.Z. supervised the research; L.J.N., N.W., D.A., X.L.Y., D.A., G.K., E.G., and S.Z. provided key reagents, methods, and technologies; N.G., L.J.N., N.W., G.V., W.C., B.K., M.E., F.M., S.W., S.B., and D.V. performed the experiments; all authors analyzed the data; all authors interpreted the results; N.G., G.V., M.E., W.C., F.M., M.M., E.G., X.L.Y., D.V., S.Z., and G.S. wrote the paper; all authors read and approved the manuscript.

## Competing interests

The authors declare no competing interests.
