## [Peer Review File · Nature Communications]

Neuropilin 1 and its ligand mini-tryptophanyl-tRNA synthetase oppositely regulate VE-cadherin turnover and vascular permeabilityREVIEWER COMMENTS

Reviewer #1 (Remarks to the Author):

The mechanisms of action of Nrp1 are still unclear, but there is a sense that it acts to regulate the surface expression of a range of important mediators of endothelial signalling and adhesion. Here the authors use an unbiased screen to identify proteins associated with Nrp1 at the cell surface and upon endocytosis. The study shows that Nrp1 can associate with VE-cadherin. Intriguing evidence for regulation of VE-cadherin internalisation by a truncated tRNA synthase are presented (mini-WARS), although further work is required to establish this.

Specific points

The authors identify many potential interaction partners of Nrp1. This is a useful resource; however, the text describing these candidate needs to be revised to make it clear that these are only candidates (there is no validation). It is likely that some of these candidates will not validate and that is fine, but this needs to be clear. I think that this section should be shortened to remove a lot of the speculation around possible functions of these potential interactions. It is better to focus on mini-WARS, which is the central story.

In Figure 2D, the authors use the STRING database to show interactions between candidate Nrp1 binding partners. Can they clarify what level of evidence they selected for this in the analysis? Was this the very broad filters (e.g. datamining) or the stronger ones (e.g. experimental evidence)?

In Figure 3C and D the authors show that VE-cadherin and PECAM can be detected in immunoprecipitates of Nrp1. As far as it is possible to say from a western blot, these do not look like strong interactions. The fact that both VE-cadherin and PECAM associate suggests that this may be general associations of Nrp1 with junctional complexes. I think that the authors minimally need to establish whether the interaction between VE-cadherin and Nrp1 is direct or indirect.

In Figure 4A-D, the authors use a single pool of 3 Nrp1 siRNAs. The pool has been used before, but that doesn't stop it having off-target effects here. There is a significant risk that any or all observations are off-target, and the authors need to repeat these experiments with a second siRNA/siRNA pool or provide rescue experiments. This is not a trivial point. Unfortunately, it also applies to the work with WARS siRNA in Figure 5.

In the quantification of the data in Figure 4A, the authors perform two independent experiments, but then pool data points for analysis; meaning that the 'n number' becomes the number of observations and that there is no analysis of reproducibility. The findings are central to the paper, and the authors need to analysis independent experiments. The same is true for the data in Figures 4C and 5D.

In Figures 4E, 5G and 5H the authors measure flux of a fluid marker across the endothelial barrier. While most of the flux will be paracellular, the assay does not distinguish between transcellular and paracellular transport, and the authors should describe this simply as a permeability assay.

In Figure 5B, the authors assay binding of Nrp1 to WARS by immunoprecipitation. They describe 3 forms of WARS in the lysate; however, the band that immunoprecipitates with Nrp1 does not migrate with any of those three. The green arrow marking the middle band is at different heights in the two lanes of the same blot. The interactions shown in Figure 5C are more convincing and support the finding.

In Figures 5G and 5H, the authors show that overexpression of mini WARS is protective against histamine-induced permeability, and the silencing of mini WARS increases the permeability response. We don't know if this is via Nrp1 however. The authors cite a previous publication showing that mini WARS interacts directly with VE-cadherin, and it seems possible that this could be a direct effect of mini WARS on VE-cadherin.

In think the authors have found an interesting new lead on Nrp1 function that would merit publication in this journal with additional work. Some significant work is required to ensure that the central findings are correct (listed above). I think the authors also need to show that mini WARS is binding to Nrp directly and that the effects of mini WARS on permeability involve Nrp1. At the moment we have no evidence that Nrp1 is binding directly to mini WARS OR VE-cadherin, whereas we already know that they interact directly with each other. It is important to be sure that this is a Nrp1 story.

Reviewer #2 (Remarks to the Author):

NRP-1 has been shown to act as a co-receptor in SEMA3A and VEGF-A signaling regulated vascular development. Additionally, NRP-1 can act as an endocytic receptor promoting integrin dependent

adhesion and vascular permeability. In this study the authors address ligands and mechanisms by which NRP1 might be involved in regulating adhesion and permeability properties of the vasculature.

The authors use a Halo-tag based pull down in combination with Mass spectrometry to identify NRP1 interactors at the plasma membrane and in the endosomal compartment. They identified putative interactors, such as cell-to-cell adhesion receptors, aaRSs, immune regulators and metabolic regulators, and surprisingly proteins involved in glycolysis.

Next the authors address NRP1 localization to intercellular contacts and its interaction with cell-to-cell adhesion receptors, like VE-cadherin and PECAM1. They use FRAP analysis to show, that VE-cadherin-mCherry mobility depends on NRP1 and that NRP1 promotes EC permeability.

Finally the authors claim that a secreted fragment of Tryptophanyl-tRNA synthetase (mini-WARS) acts as a NRP1 inhibitory ligand that impairs the internalization of NRP1 and its interactor VE-cadherin and thereby regulates adherence junctions and vascular permeability.

The manuscript combines a number of interesting findings, which are of interest to the vascular community.

However there are a number of obstacles to publishing the manuscript in its current form:

The manuscript is generally difficult to read, which seems to me partially based on lack of focus on one specific function. Especially the proteomics approach gives rise to interesting aspects of NRP-1 biology, but completely disconnects the reader from the title story. I think it should be subject of intense discussion with the editorial team on whether or how to include this data.

Placing experimental descriptions like the use of the Halo-tag in the introduction is also disruptive to an easy read. Instead the discussion should be more focussed, and contain more data on NRP1 biology, and the already described interactions with other receptors as well as known ligands.

Likewise the multiple discussion of CMT2D GARS mutations and their disease function is completely out of the scope of the manuscript. It would be much better to focus on the actual data, than to speculate to that extend on similarities, without providing data.

The interaction with Aminoacyl-tRNA synthetases remains unclear: on the one hand the authors claim to have found GARS, HARS and WARS as NRP1 interactors, on the other hand they claim that all of these share a WHEP domain which inhibits interaction with NRP1, and only mini-WARS can interact with NRP1. This needs clarification.

The authors also need to provide more insight into the processing or productions and especially secretion of mini-WARS. It is of profound importance how the players NRP1 and mini-WARS get to interact in the extracellular space.

Fig1 C, D: the purpose of the images is less clear, it would help if the important message of the figure would not be hidden somewhere in the middle (“HaloTag-Alexa660 bound HT-NRP1 localized on the surface of ECs kept at 4°C (C) and is efficiently endocytosed only upon incubation at 37°C for 3 min, where HT-NRP1 co-localized in EEA1-positive early endosomes)“

This relocalisation should be indicated by arrows etc...also for colocalization a different colour scheme might improve matters drastically as one can not observe any colocalization in the merged image. It is also difficult to understand what the pCCLHT-NRP1 GFP channel is supposed to add? The image would benefit from its removal

HaloTag-Alexa660 bound HT-NRP1 localized on the surface of ECs kept at 4°C (C) and is efficiently endocytosed only upon incubation at 37°C for 3 min, where HT-NRP1 co-localized in EEA1-positive early endosomes

Fig1 F: HT-NRP1 rescues the defective adhesion of siNRP1 ECs to fibronectin.

Shown is only a quantification of adherent cells, what happened to the others? Could they also have died? Did you quantify non-adherent cells in the supernatant to ensure viability was not affected, and only adhesion is compromised?

It seems surprising that NRP1 should be solely required for adhesion, and lacks a mechanism, as HUVECs deprived of VE-Cadherin do not fail to adhere. As a control VE-cadherin deficiency should be included, as well as other mediators of this supposedly NRP1 dependent adhesion. Western blot and immunohistochemistry for NRP1, VE-cadherin and PECAM should be provided at least!

Also, in the following figures , e.g. Figure 4 siNRP did not disturb confluency of endothelial cell patches, and lead to increased adherence junction VE-cadherin intensity.

The manuscript does not compare these effects.

Fig2 A: „Schematic drawing summarizing the role of NRP1 in the endosomal trafficking of plasma membrane receptors“.

This drawing does only depict the presence of NRP1 in endosomal vesicles, but does by no means summarize the role of NRP1. Without any further explanation this drawing only indicates, that somehow alpha and beta integrins move from the early endosome back to the cell surface. Not helpful for this paper.

Fig 4: the fact that lack of NRP1 impairs VE-cadherin fluorescence recovery after bleaching and keeps VE-Cadherin fluorescence intensity at the cell-cell contact high, does not imply that NRP1 directly promotes VE-cadherin turnover! Please be careful not to overinterpret results. Lack of NRP1 could generally slow down vesicle/endosomal trafficking, and even that could be an indirect effect.

To show a more direct link, depletion of other endosomal cycling receptors should not show the same effect...

Likewise: does depletion of VegfR2 have similar effects on VE-cadherin? How are the effects on VE-cadherin of e.g. endosomal rabs (rab4, rab5, rab7) are depleted? Or alternatively: how would the authors propose to show that this is specific to a NRP1-VEcadherin interaction and not a general effect?

Fig4F it is very commendable that there is at least one experiment linking the observed data to an in vivo function. However, while MECA32 staining was done, it was not quantified. On the representative picture the number of MECA32 positive ECs seem reduced, which would strongly bias the results of Evans blue leakage per dried skin weight. It would not be surprising to observe less leakage via fewer vessels. Therefore quantification of ECs per analysed skin patch is crucial.

Fig 5D: the authors continuously make it difficult for the reader to follow their reasoning. This is a good example. It would greatly improve the manuscript if a clear statement of the figure result was included, ideally before the detailed description of the experiment. E.g. this figure might read: "overexpression of mini-WARS increases NRP1 and VE-cadherin expression at cell-cell contacts"

Fig 5 F the graph labels are confusing: "percentage of surface 5' internalized NRP1"

I assume the numbers 5' and 1' are supposed to read:"to" or "over"??

Fig 6 : while Figure 6 attempts to illustrate a model, it fails to be instructive beyond the figure legend. Whereas the goal of a model illustration should be to visualise a process, the shown illustrations are beautiful, but fail to convey any message without reading the full figure legend. Ideally an illustration would visualise the concept without a full written explanation necessary.

Reviewer #3 (Remarks to the Author):

In their manuscript entitled "Neuropilin 1 and its ligand mini-tryptophanyl-tRNA synthetase oppositely regulate VE-cadherin turnover and vascular permeability" by Gioelli et al., the authors report that NRP1 acts as an endocytic chaperone primarily for adhesion receptors and interacts with VE-cadherin promoting its basal internalization-dependent turnover, both in vitro and in vivo. They further show that a splice variant of tryptophanyl-tRNA synthetase (mini-WARS) act as a unconventional extracellular inhibitory ligand of NRP1 that stabilizes NRP1 at the adherens junctions, and slows down both VE-cadherin turnover and histamine-elicited endothelial leakage. Overall, this is an

interesting small study, however in the current form somewhat preliminary in its findings. Also, a short version would be much more appropriate with the amount of current data in the manuscript.

Minor quibbles:

-The authors identify several proteins interacting with the NRP1 using the HaloTag IP and MS analysis. However, the filtering strategy to identify the high-confidence interactors is not performed in the current state-of-the-art fashion. At least the authors should compare their data with the CRAPome (www.crapome.org) interaction proteomics contaminant database and see with what frequency and amounts their suggested NRP1 interactors are detected in the CRAPome.

-Control would be better abbreviated as “ctrl”

-scale bars are somewhat exotic (e.g. 6 and 13.6 μm)

-scale bar missing in the middle panels of Fig4. A

-it is difficult to understand how some of the *** p-values were obtained in the Figure 4 C and thereafter. The values from these experiments should be listed in a supplementary table as well.

-Figure 6. is not very informative and would be better justified in the supplement.

Referee #1:

The mechanisms of action of Nrp1 are still unclear, but there is a sense that it acts to regulate the surface expression of a range of important mediators of endothelial signalling and adhesion. Here the authors use an unbiased screen to identify proteins associated with Nrp1 at the cell surface and upon endocytosis. The study shows that Nrp1 can associate with VE-cadherin. Intriguing evidence for regulation of VE-cadherin internalisation by a truncated tRNAsynthase are presented (mini-WARS), although further work is required to establish this.

We are glad that the reviewer found our work of interest and thank her/him for the constructive criticisms and comments.

Specific points:

1. The authors identify many potential interaction partners of Nrp1. This is a useful resource; however, the text describing these candidate needs to be revised to make it clear that these are only candidates (there is no validation). It is likely that some of these candidates will not validate and that is fine, but this needs to be clear. I think that this section should be shortened to remove a lot of the speculation around possible functions of these potential interactions. It is better to focus on mini-WARS, which is the central story.

We modified the **Results section (pages 6-9)** that describes the proteins identified by shotgun MS analysis to co-immunoprecipitate with NRP1. We made clear that **most of the identified NRP1 interaction partners are candidates** (“putative direct or indirect interaction partners of NRP1”, “To understand the roles that NRP1 may play”), while few of them (e.g. integrins) confirm previous hypothesis-driven interactions that were subsequently experimentally validated and confirmed (“In sum, our proteomic analysis has identified established interactors of NRP1, but also a plethora of novel putative interactors”). We also **shortened by about 50%** the section on **pages 8-9, removed many speculations**, and left just essential descriptions of potential candidate interactors.

2. In Figure 2D, the authors use the STRING database to show interactions between candidate Nrp1 binding partners. Can they clarify what level of evidence they selected for this in the analysis? Was this the very broad filters (e.g. datamining) or the stronger ones (e.g. experimental evidence)?

This information is included in the **Materials and Methods section (page 27)**: “The physical and functional interactions of the networks generated with STRING were defined based on **default parameters, with minimum required confidence score of 0.400, but** using only **“Neighborhood”, “Experiments” and “Databases” as active interaction sources**”.

3. In Figure 3C and D the authors show that VE-cadherin and PECAM can be detected in immunoprecipitates of Nrp1. As far as it is possible to say from a western blot, these do not look like strong interactions. The fact that both VE-cadherin and PECAM associate suggests that this may be general associations of Nrp1 with junctional complexes. I think that the authors minimally need to establish whether the interaction between VE-cadherin and Nrp1 is direct or indirect.

As shown in **Figure 3E** and described **on page 10**, we now provide formal biochemical evidence that, in a pull-down assay employing agarose beads conjugated to an anti-NRP1 antibody, the purified recombinant whole extracellular portions of **VE-cadherin and NRP1 can directly interact**.

4. In Figure 4A-D, the authors use a single pool of 3 Nrp1 siRNAs. The pool has been used before, but that doesn't stop it having off-target effects here. There is a significant risk that any or all observations are off-target, and the authors need to repeat these experiments with a second siRNA/siRNA pool or provide rescue experiments. This is not a trivial point. Unfortunately, it also applies to the work with WARS siRNA in Figure 5.

We **repeated** experiments by silencing NRP1 in human endothelial cells (**sihNRP1**) and **rescuing** its functions by retroviral delivery of **silencing resistant** HA-tagged wild type mouse Nrp1 cDNA (**mNrp1**), as previously described (Valdembri et al., 2009, *PLoS Biol.* 7(1): e1000025. doi: 10.1371/journal.pbio.1000025). We found that **mNrp1 successfully rescued** the following abnormal **phenotypes caused by human NRP1 silencing**:

- i) **larger VE-cadherin-containing** intercellular adherens junctions (**AJs**), as evaluated by quantitative fluorescence confocal microscopy (**Figure 4A**);
- ii) **decreased VE-cadherin endocytosis** from the cell surface, as evaluated by biochemical internalization assays (**Figure 4D**);
- iii) **decreased** histamine-elicited **vascular permeability** (**Figure 4F**).

We also **repeated** experiments in which, by means of oligonucleotides targeting the 3' non-translated portion of its mRNA, we **silenced** WARS in human endothelial cells (**siWARS**) and **selectively rescued** mini-WARS functions by lentiviral delivery of **silencing resistant** wild type human **mini-WARS** cDNA. We found that indeed silencing resistant **mini-WARS successfully rescues** the **increased** histamine-elicited **vascular permeability phenotype caused by WARS silencing** (**Figure 6B**).

5. In the quantification of the data in Figure 4A, the authors perform two independent experiments, but then pool datapoints for analysis; meaning that the 'n number' becomes the number of observations and that there is no analysis of reproducibility. The findings are central to the paper, and the authors need to analysis independent experiments. The same is true for the data in Figures 4C and 5D.

For each experiment of **new Figures 4A, 4C, 4D, 4E, 4F, 5F, 5H, and 6A-C**, we performed **three independent biological replicates**, in which at least three technical replicates were performed. Then, to analyze the reproducibility, we calculated **the mean of the three independent biological replicates \pm SEM** and performed statistical analyses.

6. In Figures 4E, 5G and 5H the authors measure flux of a fluid marker across the endothelial barrier. While most of the flux will be paracellular, the assay does not distinguish between transcellular and paracellular transport, and the authors should describe this simply as a permeability assay.

We now describe the experiments of **Figures 4F (former Fig. 4E) and 6A-C (former 5G and H)** as **permeability assays**.

7. In Figure 5B, the authors assay binding of Nrp1 to WARS by immunoprecipitation. They describe 3 forms of WARS in the lysate; however, the band that immunoprecipitates with Nrp1 does not migrate with any of those three. The green arrow marking the middle band is at different heights in the two lanes of the same blot. The interactions shown in Figure 5C are more convincing and support the finding.

To analyze the interaction of NRP1 and the three different WARS isoforms further and better, we verified the ability of **recombinant purified Fc-tagged extracellular portion of NRP1** to interact with **recombinant purified 6xHis-tagged full length (FL), or mini- or T2-WARS**. As shown in **Figures S2B and 5C** and described **on page 13**, in agreement with our experiments in endothelial cells, *in vitro* pull-down assays on Protein G beads clearly showed that the purified Fc-tagged extracellular portion of **NRP1 interacts preferentially with purified mini-WARS**, compared with FL or T2-WARS.

8. In Figures 5G and 5H, the authors show that overexpression of mini WARS is protective against histamine-induced permeability, and the silencing of mini WARS increases the permeability response. We don't know if this is via Nrp1 however. The authors cite a previous publication showing that mini WARS interacts directly with VE-cadherin, and it seems possible that this could be a direct effect of mini WARS on VE-cadherin.

In think the authors have found an interesting new lead on Nrp1 function that would merit publication in this journal with additional work. Some significant work is required to ensure that the central findings are correct (listed above). I think the authors also need to show that mini WARS is binding to Nrp directly and that the effects of mini WARS on permeability involve Nrp1. At the moment we have no evidence that Nrp1 is binding directly to mini WARS OR VE-cadherin, whereas we already know that they interact directly with each other. It is important to be sure that this is a Nrp1 story.

As mentioned in **our reply to point #7**, we showed that purified recombinant **mini-WARS directly binds with high affinity** to purified recombinant extracellular portion of **NRP1**. Next, we directly assessed whether the overexpression of mini-WARS is protective against histamine-induced permeability *via* NRP1. To this aim, as shown in **Figure 6C** and described on **page 15**, we compared the impact of NRP1 silencing on the ability of overexpressed mini-WARS to impair histamine-elicited permeability in endothelial cells. We clearly observed that, **upon NRP1 silencing, mini-WARS overexpression is no longer able** to hinder histamine-elicited permeability (**Figure 6C**). Hence, we conclude that to exert its **inhibitory effect on endothelial permeability mini-WARS requires NRP1**, the presence of **VE-cadherin alone not being sufficient** in this regard. Altogether, our new set of experiments provide direct evidence that mini-WARS binds to NRP1 directly and that the effect of mini-WARS on permeability involves NRP1.

Referee #2:

NRP-1 has been shown to act as a co-receptor in SEMA3A and VEGF-A signaling regulated vascular development. Additionally, NRP-1 can act as an endocytic receptor promoting integrin dependent adhesion and vascular permeability. In this study the authors address ligands and mechanisms by which NRP1 might be involved in regulating adhesion and permeability properties of the vasculature.

The authors use a Halo-tag based pull down in combination with Mass spectrometry to identify NRP1 interactors at the plasma membrane and in the endosomal compartment. They identified putative interactors, such as cell-to-cell adhesion receptors, aARs, immune regulators and metabolic regulators, and surprisingly proteins involved in glycolysis.

Next the authors address NRP1 localization to intercellular contacts and its interaction with cell-to-cell adhesion receptors, like VE-cadherin and PECAM1. They use FRAP analysis to show, that VE-cadherin-mCherry mobility depends on NRP1 and that NRP1 promotes EC permeability.

Finally the authors claim that a secreted fragment of Tryptophanyl-tRNA synthetase (mini-WARS) acts as a NRP1 inhibitory ligand that impairs the internalization of NRP1 and its interactor VE-cadherin and thereby regulates adherence junctions and vascular permeability.

The manuscript combines a number of interesting findings, which are of interest to the vascular community.

We are glad that the reviewer found our work of interest and thank her/him for the constructive criticisms and comments.

However there are a number of obstacles to publishing the manuscript in its current form:

1. The manuscript is generally difficult to read, which seems to me partially based on lack of focus on one specific function. Especially the proteomics approach gives rise to interesting aspects of NRP-1 biology, but completely disconnects the reader from the title story. I think it should be subject of intense discussion with the editorial team on whether or how to include this data.

Placing experimental descriptions like the use of the Halo-tag in the introduction is also disruptive to an easy read. Instead the discussion should be more focussed, and contain more data on NRP1 biology, and the already described interactions with other receptors as well as known ligands.

Likewise, the multiple discussion of CMT2D GARS mutations and their disease function is completely out of the scope of the manuscript. It would be much better to focus on the actual data, than to speculate to that extend on similarities, without providing data.

To make the manuscript reading easier, we modified the **Results section (pages 6-9)** that describes the proteins identified by shotgun MS analysis to co-immunoprecipitate with NRP1. We made clear that **most of the identified NRP1 interaction partners are candidates** (“putative direct or indirect interaction partners of NRP1”, “To understand the roles that NRP1 may play”), while few of them (e.g. integrins) confirm previous hypothesis-driven interactions that were then experimentally confirmed (“In sum, our proteomic analysis has identified established interactors of NRP1, but also a plethora of novel putative interactors”). We also **shortened by about 40%** the section on **pages 8-9, removed many speculations**, and left just essential descriptions of potential candidate interactors.

We also **removed** the description of the use of **the Halo-Tag** from the **Introduction (page 3-5)** and moved it in the **Results** section (**page 6**).

In the **Discussion**, we **shortened** the part concerning **CMT2D GARS mutations (page 18-19)** and we **focused more on** the role of **mini-WARS/NRP1 interaction and function**.

2. *The interaction with Aminoacyl-tRNA synthetases remains unclear: on the one hand the authors claim to have found GARS, HARS and WARS as NRP1 interactors, on the other hand they claim that all of these share a WHEP domain which inhibits interaction with NRP1, and only mini-WARS can interact with NRP1. This needs clarification.*

The authors also need to provide more insight into the processing or productions and especially secretion of mini-WARS. It is of profound importance how the players NRP1 and mini-WARS get to interact in the extracellular space.

Based on our previous findings on CMT2D GARS mutants (He et al., *Nature*, 2015, 526:710-714) and WARS data shown in this manuscript, it emerges that the functional inactivation (He et al., *Nature*, 2015, 526:710-714) or the physiological splicing out (this manuscript) of the WHEP domain respectively favor the high affinity binding of secreted GARS and WARS to the extracellular b1/b2 domains of NRP1. **As we discuss on page 17-18**, since, GARS, HARS and WARS are the only WHEP domain containing aaRS and the **alternative splicing-dependent removal of the WHEP domain** correlate with the ability of **mini-WARS** to bind with high affinity to NRP1, it is possible that alternatively spliced WHEP domain-lacking isoforms of GARS and HARS exist to bind and regulate NRP1 function as well. In this regard, the **alternative splicing-dependent control of WHEP inclusion or exclusion** from WARS, GARS, and HARS may **inhibit or promote** their **binding to NRP1** and control NRP1 **functions**.

Because of aminoacid motifs acquired during evolution, several aminoacyl-tRNA synthetase (**aaRS**) **proteins can act as secreted extracellular cytokines** signaling through plasma membrane receptors, which are still under scrutiny (Guo & Schimmel, 2013, *Nat. Chem. Biol.* 9, 145-153). Cytosolic AaRS are **released in the extracellular space via unconventional pathways** (Kapoor et al., 2008, *J. Biol. Chem.*, 283:2070-2077; Rabouille, 2017, *Trends Cell Biol.*, 27, 230-240). Unconventional secretion of leaderless (devoid of signal peptide) cytosolic proteins requires the formation of a large multiprotein release complex comprising several other leaderless proteins, such as **S100A10** and annexin A2 proteins (Rabouille, 2017, *Trends Cell Biol.*, 27, 230-240). Previous studies, which actively involved one of the senior authors of our manuscript (XLY), showed that the S100A10-annexin A2 complex **controls FL WARS secretion in endothelial cells**, with S100A10 acting as a negative regulator of the extracellular release of FL WARS (Kapoor et al., 2008, *J. Biol. Chem.*, 283:2070-2077). As shown in **Figure 5E** and described **on page 14**, we now provide evidence that **S100A10 also inhibits the unconventional secretion of mini-WARS in endothelial cells**. Therefore, it appears that both FL and mini-WARS rely on an S100A10-regulated unconventional mechanism to be released in the extracellular environment that surrounds endothelial cells.

3. *Fig1 C, D: the purpose of the images is less clear, it would help if the important message of the figure would not be hidden somewhere in the middle ("HaloTag-Alexa660 bound HT-NRP1 localized on the surface of ECs kept at 4°C(C) and is efficiently endocytosed only upon incubation at 37°C for 3 min, where HT-NRP1 co-localized in EEA1-positive early endosomes)"*

This relocation should be indicated by arrows etc...also for colocalization a different colour scheme might improve matters drastically as one can not observe any colocalization in the merged image. It is also difficult to understand what the pCCLHT-NRP1 GFP channel is supposed to add? The image would benefit from its removal HaloTag-Alexa660 bound HT-NRP1 localized on the surface of ECs kept at 4°C (C) and is efficiently endocytosed only upon incubation at 37°C for 3 min, where HT-NRP1 co-localized in EEA1-positive early endosomes.

In **Figure 1A-D**, we first draw the N-terminal localization, just after the signal peptide, of the HaloTag (HT) in the structure of HT-NRP1 (**Figure 1A**). Then, we demonstrate that, when delivered through the third-generation lentiviral vector pCCL.sin.cPPT.PGK.GFP.WPRE, the HT-NRP1 construct is effectively expressed in endothelial cells for at least 10 days (**Figure 1B**). Afterwards, we show how, when incubated on HT NRP1-transduced endothelial cells kept at 4°C (to inhibit endocytosis), the membrane impermeable HaloTag-Alexa660 ligand labels the surface of effectively transduced by the lentivirus (**Figure 1C**). When cells are shifted to 37 °C for 3 min, HaloTag-Alexa660 ligand labels the HT NRP1 that is internalized from the cell

surface into EEA1⁺ early endosomes (**Figure 1D**).

As described in refs. 92 and 93 (**mentioned in Materials and Methods, page 20**), the pCCL.sin.cPPT.PGK.GFP.WPRE lentivirus contains a **bidirectional promoter** that allows the simultaneous expression of both **HT-NRP1** and **GFP**, which were both imaged in **Figure 1C-D**. Following the Referee's suggestion, **we removed GFP** to allow to observe the colocalization more easily between HaloTag-Alexa660 ligand-bound HT NRP1 and the early endosome marker EEA1. Concerning the **color scheme**, we employed a palette to favor **color blind persons** following the policy of *Nature Communications* ["Authors are encouraged to consider the needs of colourblind readers (a substantial minority of the male population) when choosing colours for figures - <https://www.nature.com/ncomms/submit/how-to-submit>]. However, to further make the observation of the colocalization easy, we maintained magenta the HaloTag-Alexa660 ligand-bound HT NRP1 pseudocolor, **but we exchanged cyan for green** as early endosome marker EEA1 pseudocolor, as recently suggested (Alla Katsnelson, "Colour me better: fixing figures for colour blindness", *Nature*, 2021, 598:224-225 - <https://www.nature.com/articles/d41586-021-02696-z>).

4. Fig1 F: HT-NRP1 rescues the defective adhesion of siNRP1 ECs to fibronectin.

Shown is only a quantification of adherent cells, what happened to the others? Could they also have died? Did you quantify non-adherent cells in the supernatant to ensure viability was not affected, and only adhesion is compromised?

It seems surprising that NRP1 should be solely required for adhesion, and lacks a mechanism, as HUVECs deprived of VE-Cadherin do not fail to adhere. As a control VE-cadherin deficiency should be included, as well as other mediators of this supposedly NRP1 dependent adhesion. Western blot and immunohistochemistry for NRP1, VE-cadherin and PECAM should be provided at least!

Also, in the following figures, e.g. Figure 4 siNRP did not disturb confluency of endothelial cell patches, and lead to increased adherence junction VE-cadherin intensity.

The manuscript does not compare these effects.

In **Figure 1F**, a typical **short term (20 min) cell-to-extracellular matrix (ECM) adhesion assay** (for a recent method paper see Varol, 2020, *Methods Mol. Biol.* 2109:209-217) employed to evaluate the ability of cells to **adhere and spread via integrin receptors on ECM proteins, such as fibronectin (FN)**, coated on 96 well plates. Differently from integrins, **cell-to-cell adhesion receptors**, such as **VE-cadherin** or **PECAM1**, are **not involved** in endothelial cell adhesion to the **ECM** (Herbert & Stainier, *Nat. Rev. Mol. Cell Biol.*, 2011, 12:551–564). **Trypan blue staining** is employed and only trypan blue-negative viable cells are counted before plating.

In agreement with the work of others (Murga et al., *Blood*, 2005, 105:1992-1999; Ellison et al., 2015, *Dis. Model Mech.*, 8:1105-1119), we previously showed that the silencing of NRP1 impairs the endocytic turnover of integrin containing adhesions that is required to allow endothelial cell spreading on fibronectin (**Valdembri et al., PLoS Biol.**, 2009, 7(1):e25; DOI: 10.1371/journal.pbio.1000025). We also demonstrated how the expression of **silencing resistant NRP1 effectively rescues** this phenotype in NRP1-silenced endothelial cells (Valdembri et al., *PloS Biol.*, 2009). Here, we show that the presence of **HT does not affect** the ability of silencing resistant HT-NRP1 to **rescue** the defective spreading and adhesion on fibronectin of endothelial cells lacking NRP1.

As shown in Figure 4 (see also point #6 below), similarly to what observed for NRP1 interacting integrins, the silencing of NRP1 also impairs the endocytosis (**Figure 4D – biochemical endocytosis assay**) and turnover (**Figure 4C – fluorescence recovery after photobleaching microscopy**) of the newly identified NRP1 interactor VE-cadherin. In agreement with its reduced internalization and turnover, VE-cadherin accumulates at the intercellular junctions of NRP1 silenced endothelial cells (**Figure 4A, confocal immunofluorescence microscopy**), while VE-cadherin protein levels are not modified (**Figure 4B – Western blot analysis**).

5. Fig2 A: „Schematic drawing summarizing the role of NRP1 in the endosomal trafficking of plasma membrane receptors“.

This drawing does only depict the presence of NRP1 in endosomal vesicles, but does by no means summarize the role of NRP1. Without any further explanation this drawing only indicates, that somehow alpha and beta integrins move from the early endosome back to the cell surface. Not helpful for this paper.

The schematic of Figure 2A summarize the knowledge about NRP1 function so far, i.e., that **NRP1 acts as a pro-endocytic receptor for transmembrane proteins such as integrins** (for reviews see Caswell et al., *Nat. Rev. Mol. Cell Biol.*, 2009, 10:843-853; Bridgewater et al., *J. Cell Sci.*, 2012, 125:3695-701; De Franceschi et al., *J. Cell Sci.*, 2015, 128: 839–852). We worked to **better and further explain these concepts** in the **text** (anticipating the citation of Figure 2A at the beginning of **page 7 in a more appropriate context**) and in **Figure 2A legend**.

6. Fig 4: *the fact that lack of NRP1 impairs VE-cadherin fluorescence recovery after bleaching and keeps VE-cadherin fluorescence intensity at the cell-cell contact high, does not imply that NRP1 directly promotes VE-cadherin turnover! Please be careful not to overinterpret results. Lack of NRP1 could generally slow down vesicle/endosomal trafficking, and evvevn that could be an indirect effect.*

To show a more direct link, depletion of other endosomal cycling receptors should not show the same effect... Likewise: does depletion of VegfR2 have similar effects on VE-cadherin? How are the effects on VE-cadherin of e.g. endosomal rabs (rab4, rab5, rab7) are depleted? Or alternatively: how would the authors propose to show that this is specific to a NRP1-VEcadherin interaction and not a general effect?

As shown in **Figure 3E** and described **on page 10**, we now provide formal biochemical evidence that, in a pull-down assay employing an anti-NRP1 antibody, the **purified recombinant** whole extracellular portions of **VE-cadherin and NRP1 interact directly**.

As shown in **Figure 4E** and described **on page 11**, for **control** purposes, we show that **NRP1 silencing does not affect the internalization rate of the transferrin receptor 1**, a paradigmatic receptor employed to study endocytosis in different cell types (Hsu et al., *Nat. Rev. Mol. Cell Biol.*, 2012, 13:323–328).

To sum up:

1. In endothelial cells **NRP1 co-immunoprecipitates with VE-cadherin**, as independently evaluated by shotgun mass spectrometry (**MS**) and **Western blot** analysis.
2. The purified **recombinant extracellular portions** of NRP1 and VE-cadherin **physically interacts** *in vitro* pull-down assay.
3. Fluorescence **confocal** microscopy shows that NRP1 and VE-cadherin definitely **colocalize at endothelial intercellular junctions**.
4. **NRP1 silencing** causes an evident **accumulation of VE-cadherin at endothelial cell-to-cell contacts**, as evaluated by fluorescence **confocal** microscopy; the expression of exogenous silencing-resistant NRP1 rescues this effect.
5. **The lack of NRP1 results in a decreased VE-cadherin turnover** quantified by a **state-of-the-art method**, such as fluorescence recovery after photobleaching (**FRAP**) microscopy on living endothelial cells (Fritzsche and Charras, 2015, *Nature Protocols*, 10:660–680; Orsenigo et al., 2012, *Nature Commun.*, 3, 1208).

6. **Biochemical** internalization assays show that **NRP1 silencing** significantly **reduces the endocytosis of VE-cadherin, but not transferrin receptor 1**; the expression of exogenous silencing-resistant NRP1 rescues this effect.
7. In both **cultured human endothelial cells** and **living mice** the **lack of NRP1 strongly impairs** histamine-elicited **vascular permeability**, a function which is known to depend on the endocytosis of VE-cadherin from endothelial intercellular junctions (Claesson-Welsh et al., *Trends Mol. Med.*, 2021, 27:314-331); the expression of exogenous silencing-resistant NRP1 rescues this effect *in vitro*.

Therefore, **we propose a working model** in which, similarly to what we previously observed for integrins at cell-to-ECM contacts (Valdembri et al., *PLoS Biol.*, 2009), NRP1 physically interacts with VE-cadherin at endothelial intercellular junctions. Here, NRP1 acts as a pro-endocytic receptor promoting VE-cadherin internalization and turnover along with vascular permeability.

7. Fig4F it is very commendable that there is at least one experiment linking the observed data to an in vivo function. However, while MECA32 staining was done, it was not quantified. On the representative picture the number of MECA32 positive ECs seem reduced, which would strongly bias the results of Evans blue leakage per dried skin weight. It would not be surprising to observe less leakage via fewer vessels. Therefore quantification of ECs per analysed skin patch is crucial.

As shown in **Figure S2A** and described **on page 12**, we quantified the amount of MECA32⁺ endothelial cells per skin patch both in wild type and endothelial *Nrp1* knock-out animals without observing any statistically significant difference.

8. Fig 5D: the authors continuously make it difficult for the reader to follow their reasoning. This is a good example. It would greatly improve the manuscript if a clear statement of the figure result was included, ideally before the detailed description of the experiment. E.g. this figure might read: "overexpression of mini-WARS increases NRP1 and VE-cadherin expression at cell-cell contacts".

We modified the title of the new Figure 5 as follows: "Mini-WARS is an extracellular inhibitory NRP1 ligand that impairs VE-cadherin turnover."

9. Fig 5 F the graph labels are confusing: "percentage of surface 5' internalized NRP1" I assume the numbers 5' and 1' are supposed to read: "to" or "over"??

Previously, we employed the **prime symbol (')** to denote minutes of time in Figure 5F graph labels. We substituted the prime symbol (') with the official **SI symbol for minute**, which is **min** (without a dot).

10. Fig 6 : while Figure 6 attempts to illustrate a model, it fails to be instructive beyond the figure legend. Whereas the goal of a model illustration should be to visualise a process, the shown illustrations are beautiful, but fail to convey any message without reading the full figure legend. Ideally an illustration would visualise the concept without a full written explanation necessary.

We agree with the Referee that without the Figure legend it may be difficult to interpret the illustrations in which we tried to summarize the working model suggested by the manuscript at our best. Therefore, we moved previous main Figure 6 in the Supplementary Material as Supplementary Figure 3.

Referee #3:

In their manuscript entitled "Neuropilin 1 and its ligand mini-tryptophanyl-tRNA synthetase oppositely regulate VE-cadherin turnover and vascular permeability" by Gioelli et al., the authors report that NRP1 acts as an endocytic chaperone primarily for adhesion receptors and interacts with VE-cadherin promoting its basal internalization-dependent turnover, both in vitro and in vivo. They further show that a splice variant of tryptophanyl-tRNA synthetase (mini-WARS) act as a unconventional extracellular inhibitory ligand of NRP1 that stabilizes NRP1 at the adherens junctions, and slows down both VE-cadherin turnover and histamine-elicited endothelial leakage. Overall, this is an interesting small study, however in the current form somewhat preliminary in its findings. Also, a short version would be much more appropriate with the amount of current data in the manuscript.

We are glad that the reviewer found our work of interest and thank her/him for the constructive criticisms and comments.

Minor quibbles:

1. The authors identify several proteins interacting with the NRP1 using the HaloTag IP and MS analysis. However, the filtering strategy to identify the high-confidence interactors is not performed in the current state-of-the-art fashion. At least the authors should compare their data with the CRAPome (www.crapome.org) interaction proteomics contaminant database and see with what frequency and amounts their suggested NRP1 interactors are detected in the CRAPome.

We made the Introduction more focused and implemented the result section.

As suggested, we downloaded the crapome generated using HaloTag for single step purification experiment in HEK293 cells from the suggested website. Of the 314 crapome proteins (sum of three experiments), 92 were among the 966 proteins quantified in our MS analysis. Of those, **only 7 (ATP5B, IPO7, PPIA, GAPDH, LDHB, TXN, RAN) were included in the NRP1 putative interactome, which comprises 114 proteins, supporting the specificity of our results for NRP1.** In **Supplementary Data Table 1** we have now added **a column called "Crapome Halo-Tag"** and highlighted the crapome proteins with a +. We have also added **a sentence at the end of the MS data analysis part of the *Materials and Methods* section:** "To exclude that the potential NRP1 interactors were proteins commonly found as background in affinity purification experiments using the Halo-Tag, we compared the list of NRP1 interactors to those found in the CRAPome of Halo-Tag affinity purification experiments (www.crapome.org). Potential background proteins have been highlighted in Supplementary Data 1."

2 Control would be better abbreviated as "ctrl" -scale bars are somewhat exotic (e.g. 6 and 13 6 μm)

Throughout the manuscript we: i) abbreviated control as "**Ctrl**"; ii) recalculated scale bars on a **decimal** basis.

3. Scale bar missing in the middle panels of Fig4. A

We added **scale bars** in in the **middle panels of Figure 4A.**

4. It is difficult to understand how some of the *** *p*-values were obtained in the Figure 4 C and thereafter. The values from these experiments should be listed in a supplementary table as well.

For each experiment of new Figures 4A, 4C, 4D, 4E, 4F, 5F, 5H and 6A-C we performed **three independent biological replicates**. For each biological replicate ≥ 3 technical replicates were performed. Then, to analyze the reproducibility, we calculated **the mean of the three independent biological replicates \pm SEM** and performed **statistical analyses**.

As per *Nature Communication* policy numerical values of raw data employed to generate all graphs are also provided within tables of separated sheets of a dedicated Supplementary Excel file (**Supplementary Data 3**).

5. Figure 6. is not very informative and would be better justified in the supplement.

We moved previous main Figure 6 in the Supplementary Material as **Supplementary Figure 3**.

REVIEWER COMMENTS

Reviewer #1 (Remarks to the Author):

The authors have put significant work into addressing the point raised in previous review. All of my previous points have been addressed satisfactorily, and without evasion. I really appreciated the very clear explanation of the changes and new work.

I have one remaining comment, and that is minor:

Point 2. I think that SMART used with these default parameters gives a fairly low quality analysis, and includes a lot of interactions that aren't physical interactions. I think that it has very limited value showing this analysis (and the analysis in Panels B and D). This Figure gives focus to the unvalidated interactions from the screen, and my sense is that a significant number of these would not necessarily validate. I think the paper would look better with this whole Figure removed or moved to Supplementary – it's much lower standard than the rest and it's not necessary. I think a simpler illustration of the key hits would be a much better replacement – and the main text does need some kind of Figure for that. I don't feel strongly enough to insist on this.

Reviewer #2 (Remarks to the Author):

The authors have sufficiently addressed the scientific concerns in the revised manuscript.

Although within the revisions it became more evident, that the paper heavily relies on the previously published work, which might reduce the novelty of the overall study.

As a minor comment I requested better titles for the figures to make the manuscript easier to read for a broader audience and gave Figure 5D as an example.

While the authors have now added a meaningful title to Fig5D, they have not adjusted others. A good new example is the new Supplemental Figure 2 (line 912 -914): Instead of writing "Blood vessel density in either Nrp1^{+/+} or Nrp1^{EC-/-} mice was evaluated ", the readers could really benefit from a title saying " Blood vessel density in either Nrp1^{+/+} or Nrp1^{EC-/-} mice showed no significant differences".

I would advise authors and editorial team to improve many of the figure descriptions to appeal to a broader audience.

Reviewer #3 (Remarks to the Author):

The authors have only partially answered the reviewers' criticism. Especially the validation of the identified candidates is missing. As the Neuropilin 1 has been identified as an additional co-receptor for SARS-CoV-2 there are several studies probing the interactions as well, however there is very little overlap between those studies and this. With the lack of proper validation it is therefore impossible to evaluate the quality of the candidate interactions presented in this manuscript. Additionally the manuscript is still too long for amount of data.

Referee #1:

The authors have put significant work into addressing the point raised in previous review. All of my previous points have been addressed satisfactorily, and without evasion. I really appreciated the very clear explanation of the changes and new work.

We are glad that the reviewer appreciated our work aimed at addressing her/his criticisms and comments. In particular, we appreciated that (s)he found that all of her/his “previous points have been addressed satisfactorily, and without evasion”.

I have one remaining comment, and that is minor:

Point 2. I think that SMART used with these default parameters gives a fairly low quality analysis, and includes a lot of interactions that aren't physical interactions. I think that it has very limited value showing this analysis (and the analysis in Panels B and D). This Figure gives focus to the unvalidated interactions from the screen, and my sense is that a significant number of these would not necessarily validate. I think the paper would look better with this whole Figure removed or moved to Supplementary – it's much lower standard than the rest and it's not necessary. I think a simpler illustration of the key hits would be a much better replacement – and the main text does need some kind of Figure for that. I don't feel strongly enough to insist on this.

Putative direct or indirect NRP1 interactors were experimentally identified by mass spectrometry of surface or endosomal NRP1 biochemically isolated from endothelial cell lysates. **STRING is an algorithm that was not used to define NRP1 interactors**, but to perform computer-assisted enrichment analysis of KEGG pathways of NRP1 interactors experimentally identified by mass spectrometry, and to visualize them with the help of a network representing potential physical and functional interactions between interactors. To avoid representing low-confidence physical/functional interactions among the putative Nrp1 interactors found in our MS-proteomic analysis, we used only the following categories “Neighborhood”, “Experiments” and “Databases” to define an interaction. Most of the interactions (physical or functional) represented in those three categories are therefore supported by experimental evidence. For this reason, we used a rather low confidence score of 0.400.

However, to satisfy the comment of this Reviewer, we have simplified Figure 2, which now contains two categories/KEGG pathways of NRP1 interacting proteins, namely transmembrane proteins (including VE-cadherin and PECAM1 in addition to previously identified integrins) and aminoacyl-tRNA biosynthesis-related proteins (including WARS/mini-WARS), that we characterized throughout the manuscript. We moved into **Supplementary Figure 2 the other four categories/KEGG pathways of putative NRP1 interacting proteins** (vesicular proteins and proteins involved in cell adhesion, infection, and metabolism). Furthermore, for sake of clarity, we moved part of the text describing the unvalidated interactors from the main body into the legends of **Figure 2** and **Supplementary Figure 2**.

Referee #2:

The authors have sufficiently addressed the scientific concerns in the revised manuscript. Although within the revisions it became more evident, that the paper heavily relies on the previously published work, which might reduce the novelty of the overall study.

We are glad that the reviewer appreciated our work aimed at addressing her/his criticisms and comments. In particular, we appreciated that (s)he found that we “have sufficiently addressed the scientific concerns in the revised manuscript”.

As a minor comment I requested better titles for the figures to make the manuscript easier to read for a broader audience and gave Figure 5D as an example. While the authors have now added a meaningful title to Fig5D, they have not adjusted others. A good new example is the new Supplemental Figure 2 (line 912 - 914): Instead of writing "Blood vessel density in either Nrp1+/+ or Nrp1EC-/- mice was evaluated ", the readers could really benefit from a title saying " Blood vessel density in either Nrp1+/+ or Nrp1EC-/- mice showed no significant differences". I would advise authors and editorial team to improve many of the figure descriptions to appeal to a broader audience.

We further worked to modify titles, subtitles, and legends of figures (pages 34-41).

Referee #3:

The authors have only partially answered the reviewers' criticism. Especially the validation of the identified candidates is missing. As the Neuropilin 1 has been identified as an additional co-receptor for SARS-CoV-2 there are several studies probing the interactions as well, however there is very little overlap between those studies and this. With the lack of proper validation it is therefore impossible to evaluate the quality of the candidate interactions presented in this manuscript. Additionally the manuscript is still too long for amount of data.

Concerning Reviewer #3 statement "The authors have only partially answered the reviewers' criticism", we respectfully disagree with such comment, since Reviewer #1 and Reviewer #2 stated the following: #1) **"All of my previous points have been addressed satisfactorily, and without evasion"**. #2) **"The authors have sufficiently addressed the scientific concerns in the revised manuscript"**.

Regarding **this Reviewer** criticisms, **in the first round of revision**, (s)he **did not mention the need for further experimental validation** of the mass spectrometry (MS) proteomic data. He/she **asked only for the following "Minor quibbles"**: The authors identify several proteins interacting with the NRP1 using the HaloTag IP and MS analysis. However, the filtering strategy to identify the high-confidence interactors is not performed in the current state-of-the-art fashion. At least the authors should compare their data with the CRAPome (www.crapome.org) interaction proteomics contaminant database and see with what frequency and amounts their suggested NRP1 interactors are detected in the CRAPome." **We have done that analysis requested by Reviewer #3 and included it in the first revised version of the manuscript (see pages 25-26).**

Regarding this Reviewer's concerns that our data did not mirror NRP1 interactors found in SARS-CoV-2 studies, **we did not find any existing published manuscript in which authors isolated NRP1 and identified its potential direct interactors by MS in endothelial cells or in any other non-endothelial cell type**. We are only aware of a single study that has been posted as a **pre-print on bioRxiv** by the laboratory of Prof. Peter Cullen that however does not show any MS analysis of NRP1 interactome (<https://www.biorxiv.org/content/10.1101/2022.01.20.477115v1>). If this is the single manuscript the reviewer thought of, we know it very well. Since Cullen and colleagues had previously identified NRP1 as a SARS-CoV-2 co-receptor, they now show that SNX5/6 mediate the trafficking to the TGN of engineered nanoparticles, functionalized with the NRP1-interacting peptide of the SARS-CoV-2 Spike protein. However, this study **analyzed the differential TGN proteome of control versus SNX5/SNX6 silenced ovarian carcinoma HeLa cells by proximity biotinylation and does not study a NRP1-associated interactome**. On the other hand, some MS studies (e.g. Liu et al., Mol. Syst. Biol., 2021, 17:e10396; 10.15252/msb.202110396) were performed to identify proteins associated with internalized SARS-CoV-2 virus, which is however well known to employ ACE2 as the main receptor rather than NRP1 and integrins, which function only as co-receptors. Therefore, **by no means SARS-CoV-2 can be employed as a proxy to identify NRP1 interactors**.

Finally, our manuscript, after a **thorough experimental characterization**, unveils the role of NRP1 in regulating vascular permeability dependent on VE-cadherin turnover and the role of mini-WARS as the firstly identified physiological NRP1 inhibitory ligand. As also requested by Reviewer #1, **we have reorganized and simplified Figure 2** and moved the other four categories/KEGG pathways of putative NRP1 interacting proteins in **Supplementary Figure 2**. Furthermore, for sake of clarity, we moved part of the text describing the unvalidated interactors from the main body into the legends of **Figure 2** and **Supplementary Figure 2**. We would also highlight that **further experimental validation of the data would not have any impact on the key message and novelty of the paper** which is that NRP1 and its inhibitory ligand mini-WARS oppositely regulate VE-cadherin turnover and vascular permeability, as stated in the title of our manuscript. Furthermore, since, similarly to Reviewer #1, Reviewer #3 stated "Additionally the manuscript is still too long for the amount of data", those experimental validation requested by Reviewer #3 in this second round of revision would further increase the length of the work and would be **patently conflicting with the reasonable request of Reviewer #1 and the statement of Reviewer #3 her/himself on the length of the manuscript**.